# A probiotic bi-functional peptidoglycan hydrolase sheds NOD2 ligands to regulate gut homeostasis in female mice

Jie Gao[1,6], Lei Wang[1,6], Jing Jiang[2,6], Qian Xu[1,6], Nianyi Zeng[1], Bingyun Lu[3], Peibo Yuan[1], Kai Sun [4] ✉, Hongwei Zhou [1,5] ✉ & Xiaolong He [1] ✉

Secreted proteins are one of the direct molecular mechanisms by which microbiota influence the host, thus constituting a promising field for drug discovery. Here, through bioinformatics-guided screening of the secretome of clinically established probiotics from *Lactobacillus*, we identify an uncharacterized secreted protein (named LPH here) that is shared by most of these probiotic strains (8/10) and demonstrate that it protects female mice from colitis in multiple models. Functional studies show that LPH is a bi-functional peptidoglycan hydrolase with both N-Acetyl-β-D-muramidase and DL-endopeptidase activities that can generate muramyl dipeptide (MDP), a NOD2 ligand. Different active site mutants of LPH in combination with *Nod2* knock-out female mice confirm that LPH exerts anti-colitis effects through MDP-NOD2 signaling. Furthermore, we validate that LPH can also exert protective effects on inflammation-associated colorectal cancer in female mice. Our study reports a probiotic enzyme that enhances NOD2 signaling in vivo in female mice and describes a molecular mechanism that may contribute to the effects of traditional *Lactobacillus* probiotics.

The beneficial effects of probiotics on gut homeostasis have long been studied, including regulating gut immunity, enhancing barrier integrity, and maintaining microbiota homeostasis[1–3]. Thus, many probiotics have shown treatment effects on gut homeostasis disturbance-related diseases in animal models, such as chemical-induced models of inflammatory bowel diseases (IBD) and colorectal cancer (CRC)[4,5]. Nevertheless, clinical studies based on the consumption of live probiotic bacteria revealed conflicting results[6–8]. This confusing situation mainly stems from two issues: inter-individual variation in colonization ability, especially in patients with dramatic changes in gut flora[9], and the unclear mechanism of probiotic action[2].

As many of the probiotic effects contributing to human health are mediated by their metabolites, a novel therapeutic strategy has emerged, which refers to metabolites secreted, degraded, or modified by the probiotics, defined as postbiotics by the International Scientific Association of Probiotics and Prebiotics[10–12]. Considering that probiotic metabolites have clear molecular structures and easily monitored bio-availabilities, this strategy has the potential to provide therapeutic efficacy while overcoming the barriers of live probiotics-based strategies[12]. However, as this field is in its infancy, the discovered effectors are only the tip of an iceberg[13–16]. Thus, identification of new probiotic effectors, and illustrating their mechanisms of protective

[1]Microbiome Medicine Center, Department of Laboratory Medicine, Zhujiang Hospital, Southern Medical University, 510655 Guangzhou, Guangdong, China. [2]Department Gerontology, Sichuan Academy of Medical Sciences & Sichuan Provincial People's Hospital, 610072 Chengdu, Sichuan, China. [3]Department of Gastroenterology, Shenzhen Hospital, Southern Medical University, 518101 Shenzhen, Guangdong, China. [4]Department of General Surgery, Nanfang Hospital, Southern Medical University, 510515 Guangzhou, Guangdong, China. [5]State Key Laboratory of Organ Failure Research, Southern Medical University, 510655 Guangzhou, Guangdong, China. [6]These authors contributed equally: Jie Gao, Lei Wang, Jing Jiang, Qian Xu. ✉e-mail: sunkai9602@sina.com; biodegradation@gmail.com; hxl2027315@smu.edu.cn

function, may not only help us understand the mechanisms of probiotic functions, but also afford new opportunities for developing precision therapeutic methods[17].

The secreted protein is one of the direct molecular mechanisms by which the bacteria interact with the host. Here, through bioinformatics-guided screening of the secretome of four *Lactobacillus* species, which have shown beneficial effects on IBD in clinical trails[18–22], a common secreted protein (named LPH here) was identified and demonstrated to alleviate colitis and CRC through shedding NOD2 ligands to regulate gut homeostasis.

## Results

### A common secreted protein from probiotics alleviates colitis in mice

We started by searching for species of *Lactobacillus* that have shown beneficial effects on IBD in clinical trials[18–22], and four species were found, namely *Lacticaseibacillus casei (L. casei)*, *Lacticaseibacillus paracasei (L. paracasei)*, *Limosilactobacillus reuteri (L. reuteri)* and *Lacticaseibacillus rhamnosus (L. rhamnosus)*. From 27119 proteins encoded by 10 strains of these species, 364 were predicted to be secreted proteins. These secreted proteins could be categorized into 159 clusters at 75% identity (Supplementary Data 1). We specially focused on one cluster, which encompassed proteins from 8/10 of these strains, covering *L. casei*, *L. paracasei*, and *L. rhamnosus* (Supplementary Data 1, Cluster 97). The representative sequence of this cluster (BAN75155.1 from *L. casei*) was a protein with an unknown function and named LPH here. The genes flanking the gene locus of LPH were also conserved in these *Lactobacillus* species, indicating the similar function of these enzymes. To validate if the LPH homologues were secreted in these species, an anti-LPH polyclonal antibody was generated using recombinant histidine-tagged LPH and immunoblot analysis showed that LPH was present both in the cell lysates and culture supernatant of these species (Fig. 1a).

We next explored whether LPH could protect mice from 2, 4, 6-trinitrobenzenesulfonic acid (TNBS)-induced colitis. Mice were administered with LPH for 3 days before inducing colitis by TNBS (Fig. 1b). To avoid the proteins being destroyed in the gastrointestinal tract, a special protein delivery method was used to deliver LPH[16], and immunofluorescence staining showed that LPH was successfully delivered to the colonic surface (Fig. 1c). Four days following colitis induction, body weight loss, colon length shortening, histopathological injury, and impaired epithelial integrity occurred (Fig. 1d–h). Pretreatment with LPH significantly alleviated these parameters (Fig. 1d–h). To further assess the protective effects of LPH on established colitis (treatment effect), we treated mice with LPH after 1 day of TNBS challenge (Fig. 1i). Likewise, LPH efficiently alleviated TNBS-induced colitis (Fig. 1j–n).

LPH's anti-colitis effects were further validated on oxazolone (OXA)- or dextran sulfate (DSS)-induced colitis models (Supplementary Fig. 1). Finally, we confirmed that the two homologues of LPH from *L. rhamnosus* (LRP) and *L. paracasei* (LPP) could also protect mice from colitis (Supplementary Fig. 2). Taken together, these results show that LPH is a common protein from several *Lactobacillus* species that exerts both preventive and therapeutic effects on multiple murine colitis models.

### Hydrolytic activity is critical for LPH's anti-colitis effects

To reveal the mechanism of LPH's colitis protective effects, we employed bioinformatics methods to analyze its sequence and structure. LPH contains a 3D-domain in its carboxyl-terminal (Fig. 2a), which always occurs in peptidoglycan hydrolases (PGHs) (Fig. 2b). To confirm the peptidoglycan hydrolytic ability of LPH, we generated a truncation mutant of LPH lacking the 3D-domain (LPH-3D). In vitro incubation of remazol-dyed peptidoglycan with LPH, but not the LPH-3D, showed hydrolysis of peptidoglycan in a dose-dependent manner (Fig. 2c).

To determine if the peptidoglycan hydrolytic function was required for LPH's colitis protective effects, mice were pretreated with LPH or LPH-3D and then induced colitis by TNBS. We found that LPH, but not LPH-3D could protect mice from colitis (Fig. 2d–h).

Then we examined whether LPH-generated products were sufficient to protect against experiment colitis, we digested commercial peptidoglycan with LPH or LPH-3D, then filtered the digests through a 5 kDa molecular weight cut-off column to exclude protein. The peptidoglycan digests of LPH, but not those of LPH-3D, protected mice from TNBS-induced colitis (Fig. 2i–m). Above all, these results indicate that the peptidoglycan hydrolytic ability is required and sufficient for LPH's colitis protective effects.

### LPH is a bi-functional peptidoglycan hydrolase

To further characterize LPH's catalytic sites, we digested purified peptidoglycan from *Micrococcus luteus (M. luteus)* or *Saphylococcus aureus (S. aureus)* with LPH, and then analyzed the digests by 8-aminonaphthalene-1,3,6-trisulfonic acid (ANTS) labeled gel electrophoresis. Surprisingly, we found that LPH could digest intact peptidoglycan to generate specific products that migrated similarly with the muramyl dipeptide (MDP), a ligand of intracellular pattern recognition receptor nucleotide-binding oligomerization domain 2 (NOD2) (Fig. 3a; Supplementary Fig. 3a, b). High performance liquid chromatography-electrospray tandem mass spectrometry (HPLC-MS/MS) and NOD2-transfected HEK293 reporter cells analysis confirmed that MDP/NOD2 ligands, are the main products of LPH (Fig. 3b; Supplementary Fig. 4a). LPH could also generate MDP from *S. aureus* or *Escherichia coli (E. coli)*, without influencing their growth (Supplementary Fig. 4b–d). Mice gavaged with LPH showed elevated NOD2 ligands in fecal extracts in a dose-dependent manner (Fig. 3c). The above results indicate that LPH can generate MDP/NOD2 ligands from a broad range of substrates with high efficiency.

To generate MDP from intact peptidoglycan, at least two enzymes are required: N-Acetyl-β-D-muramidase (cleavage the β−1,4 glycoside), and DL-endopeptidase (cleave the peptide bond between D-glutamate and L-lysine) (Fig. 3d; Supplementary Fig. 4e). This indicates that LPH may be a bi-functional PGH. To test this hypothesis, we predicted the secondary structure of LPH (Supplementary Fig. 5), and used protein homologue modeling to reveal the three-dimensional structure of the 3D-domain (Fig. 3e). We found that the 3D-domain is composed of helix and sheet (Supplementary Fig. 5), and two conserved aspartic acids (D316, D329) were located close to each other, in a space composed of negatively charged amino acids, indicating the substrate binding property of this area (Fig. 3e, left panel). Then we generated the active site mutants of D316A (LPH-AS1) or D329A (LPH-AS2). Interestingly, LPH-AS1 and LPH-AS2 lost their ability to generate MDP from intact peptidoglycan, while preserving the ability to generate MDP from peptidoglycan predigested with mutanolysin (M, a kind of N-Acetyl-β-D-muramidase) (Fig. 3b, f, g; Supplementary Fig. 3c–f, 4f). The above data reveal mutations of D316 and D329 only affect LPH's N-Acetyl-β-D-muramidase activity while preserving its DL-endopeptidase function.

To identify the active sites responsible for LPH's DL-endopeptidase function, we generated mutants of three conserved sites at the opposite side of D316 and D329 in the 3D-domain (P295A/G297A/T298A, named LPH-AS3) (Fig. 3e, right panel). Analysis of the LPH-AS3 peptidoglycan digests showed that in contrast to SagA, a typical DL-endopeptidase[23], LPH-AS3 couldn't generate MDP from mutanolysin predigested peptidoglycan, indicating it lost the DL-endopeptidase activity (Fig. 3h, left panel). To test if LPH-AS3 preserved the N-Acetyl-β-D-muramidase function, we digested peptidoglycan with LPH-AS3 and LPH-AS1/LPH-AS2. Results showed that LPH-AS1 or LPH-AS2 could generate MDP from LPH-AS3 predigested peptidoglycan (Fig. 3h, right panel;

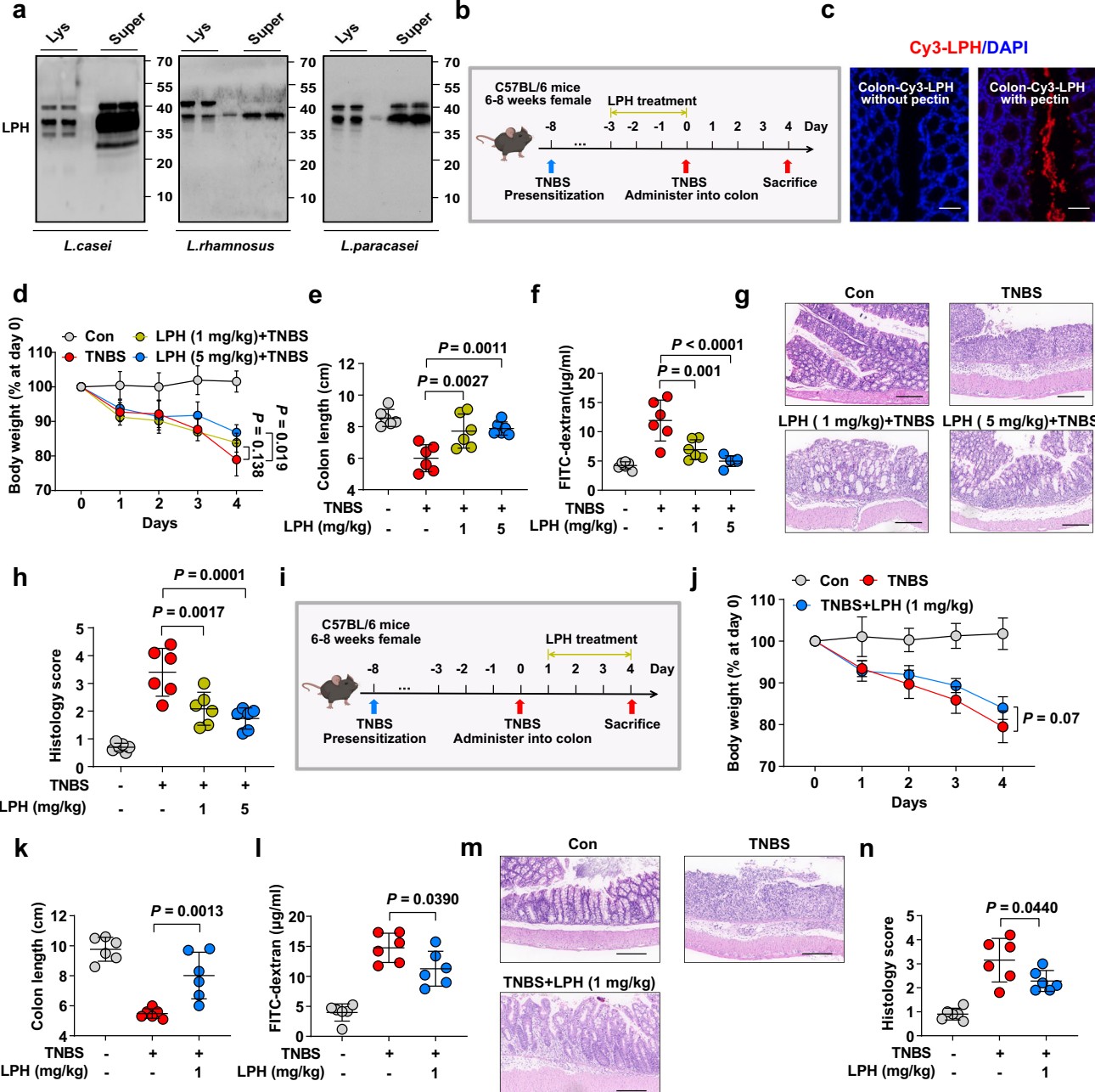

**Fig. 1 | The preventive and treatment effects of LPH against TNBS-induced colitis. a** Polyclonal antibody against LPH was generated and employed to detect LPH homologues in bacterial lysates or culture supernatants through immunoblot. Lys: bacterial lysates, Super: culture supernatant. **b** Schematic of TNBS-induced colitis and LPH treatment. C57BL/6 mice (female, *n* = 6 per group) were gavaged with pectin/zein beads containing BSA (5 mg/kg body weight) or LPH (1 mg/kg or 5 mg/kg body weight) 3 days before TNBS challenge. Mice were sacrificed 4 days after TNBS treatment. Images were created using BioRender.com. **c** Immunofluorescence staining showed LPH was successfully delivered to the colonic surface (scale bar, 40 μm). LPH (red) and nucleus (DAPI; blue). **d-j** The preventive effects of LPH on TNBS-induced colitis: body weight loss (**d**); colon length (**e**); serum FITC-dextran levels (**f**); representative H&E staining of colonic tissue, scale bar, 200 μm (**g**); the semiquantitative scoring of inflammation (**h**);

**i** Schematic of TNBS-induced colitis and LPH treatment. C57BL/6 mice (female, *n* = 6 per group) were induced colitis by TNBS on day 0 and then treated with pectin/zein beads containing BSA or LPH (1 mg/kg body weight) from days 1-4. Images were created using BioRender.com. **j-n** The treatment effects of LPH on TNBS-induced colitis: body weight loss (**j**); colon length (**k**); serum FITC-dextran levels (**l**); representative H&E staining of colonic tissue, scale bar, 200 μm (**m**); and the semiquantitative scoring of inflammation (**n**) of mice from indicated groups. Data are representative of 3 independent experiments and presented as mean ± SD. Each dot indicates an individual mouse (*n* = 6). Statistical analyses were performed using repeated measures ANOVA with Bonferroni post hoc test (**d**, **j**), one-way ANOVA with Bonferroni post hoc test (**e**, **f**, **h**, **k**, **l**, **n**). Source data are provided as a Source data file.

Supplementary Fig. 4g). These results reveal that P295/G297/T298 acts as the active sites for LPH's DL-endopeptidase function while having no relationship with its N-Acetyl-β-D-muramidase activity.

All the above results show that LPH is a bi-functional PGH with both the N-Acetyl-β-D-muramidase and DL-endopeptidase activity.

## MDP mediates the colitis protective effects of LPH

To further investigate the role of DL-endopeptidase and N-Acetyl-β-D-muramidase activities in LPH's colitis protective effects, we treated colitis mice with LPH, LPH-AS1/LPH-AS2 (retain DL-endopeptidase activity), or LPH-AS3 (retain N-Acetyl-β-D-muramidase activity). The

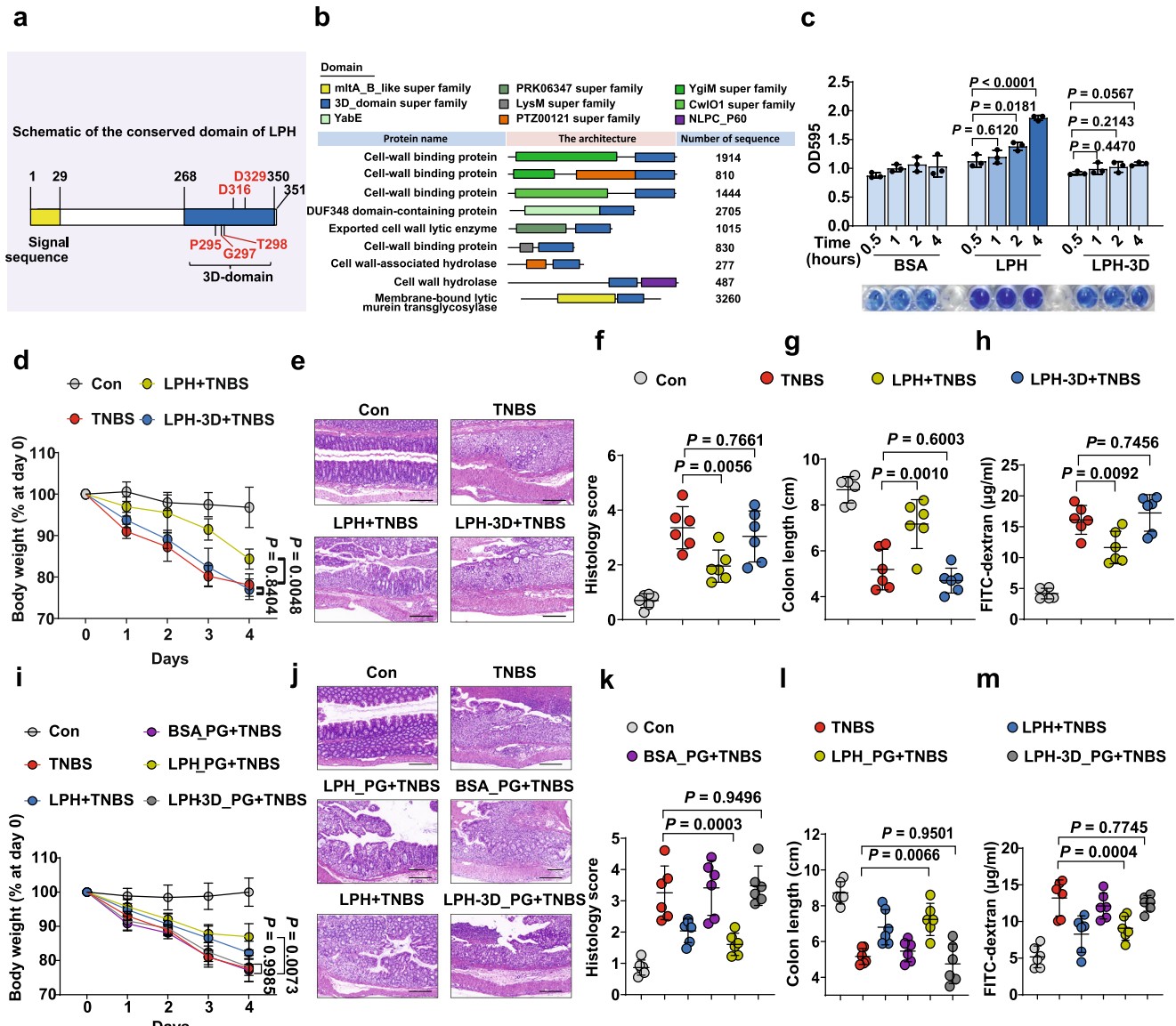

**Fig. 2 | The peptidoglycan hydrolytic ability is required and sufficient for LPH's colitis protective effects. a** Schematic of LPH's sequence organization: yellow represents the signal sequence; blue represents the conserved 3D-domain. The predicted active site residues are typed in red. **b** A summarization of proteins with the 3D-domain. **c** The ability of LPH or LPH-3D to hydrolyze remazol-dyed peptidoglycan was measured by optical density (*n* = 4 per group), data are representative of 2 independent experiments. **d**–**h** The protective effects of LPH or LPH-3D on TNBS-induced colitis, *n* = 6 (female) per group: body weight loss (**d**); representative H&E staining of colonic tissue, scale bar, 200 μm (**e**); the semiquantitative scoring of inflammation (**f**); colon length (**g**) and serum FITC-dextran level (**h**);

**i**–**m** Protective effects of peptidoglycan digests of BSA (BSA_PG), LPH (LPH_PG), or LPH-3D (LPH-3D_PG) on TNBS-induced colitis, *n* = 6 (female) per group: body weight loss (**i**); representative H&E staining of colonic tissue, scale bar, 200 μm (**j**); the semiquantitative scoring of inflammation (**k**); colon length (**l**) and serum FITC-dextran level (**m**). BSA: bovine serum albumin; LPH-3D: the 3D-domain truncation of LPH; PG: peptidoglycan from *S. aureus*. Data are representative of 3 independent experiments and presented as mean ± SD. Each dot indicates an individual mouse. Statistical analyses were performed using one-way ANOVA with Bonferroni post hoc test (**c**, **f**, **g**, **h**, **k**, **l**, **m**), repeated measures ANOVA with Bonferroni post hoc test (**d**, **i**). Source data are provided as a Source data file.

results showed that at a lower dose (1 mg/kg body weight), none of the LPH-AS1, LPH-AS2, or LPH-AS3 could protect mice from colitis (Supplementary Fig. 6a–e). Then we tested if these results were consistent at a higher dose of enzymes (5 mg/kg body weight). Surprisingly, both LPH-AS1 and LPH-AS2, but not LPH-AS3, exerted colitis protective effects at a higher dose (Fig. 4a–e). These results reveal that the DL-endopeptidase activities of LPH is critical to alleviate colitis in mice.

To analyze if the colitis-protective effects were associated with the ability to generate NOD2 ligands in vivo, we detected NOD2 ligands after administration of LPH or its mutants at different dosages using HEK293 NOD2 reporter cells. At a lower dose (1 mg/kg body weight), only LPH could elevate NOD2 ligands in the colon, ileum homogenate, and fecal extract (Fig. 4f–h). Interestingly, LPH

attenuated the abnormal elevation of NOD2 ligands in serum induced by TNBS (Fig. 4i). At a higher dose (5 mg/kg body weight), LPH-AS1 and LPH-AS2, but not LPH-AS3 could also specifically elevate NOD2 ligands in the gut and attenuate the abnormal elevation of NOD2 ligands in serum induced by TNBS (Fig. 4j–m). These results indicate the association between LPH's colitis protective effects and its ability to shed NOD2 ligands in the gut. Moreover, we validated that MDP, a well-established NOD2 ligand, displayed equivalent protective effects against TNBS-induced colitis at a dosage of 5 mg/kg in comparison to LPH at 2.5 mg/kg, which correspondingly resulted in analogous elevation of NOD2 ligands in both fecal and colonic samples (Supplementary Fig. 6f–l). These findings are consistent with our recent study[24], which demonstrated that N-Acetyl-β-D-muramidase is a

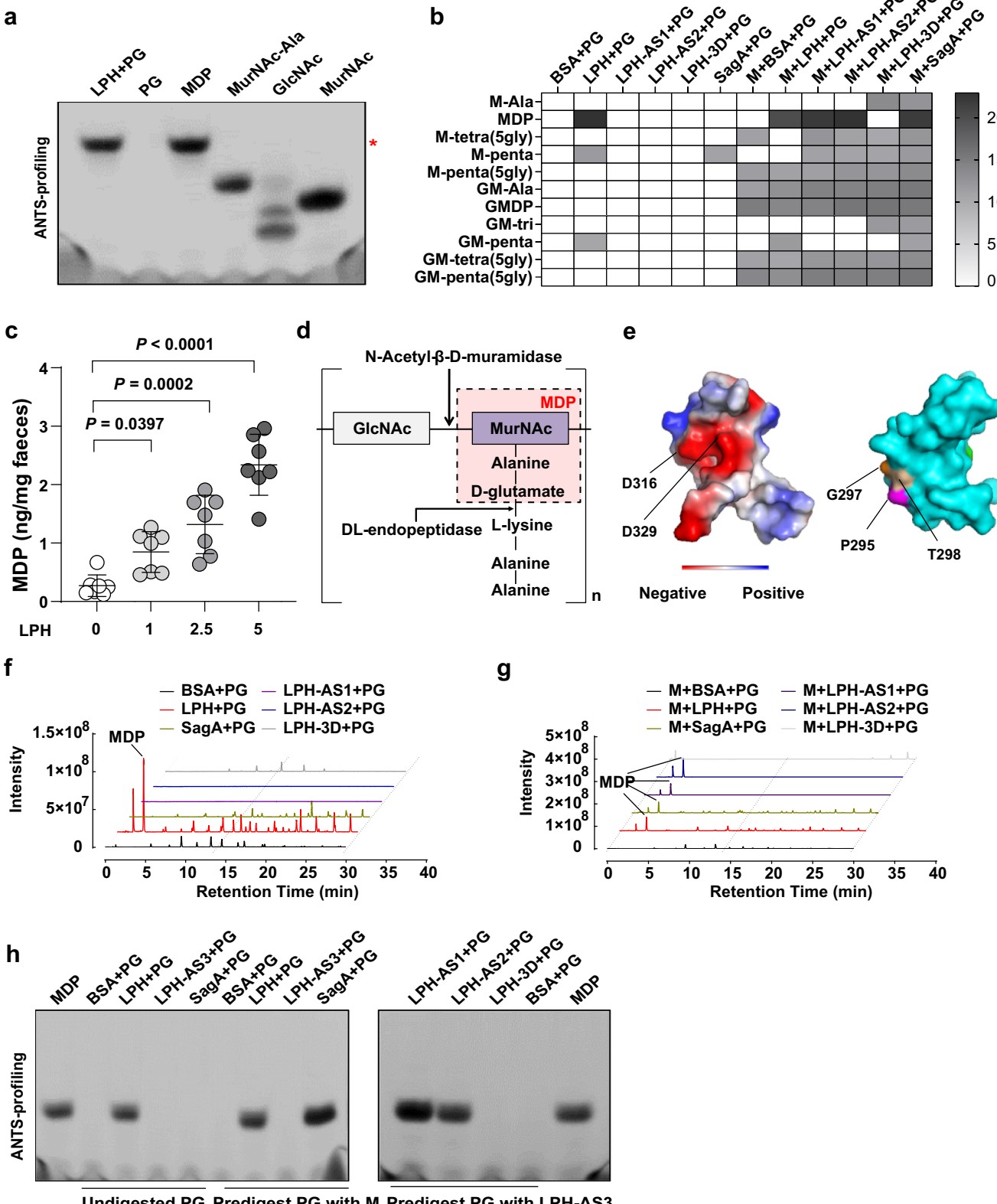

redundant enzyme in the gut microbiome, while the abundance of DL-endopeptidase is relatively lower than N-Acetyl-β-D-muramidase. Thus, supplementation of DL-endopeptidase (LPH-AS1/LPH-AS2), instead of N-Acetyl-β-D-muramidase (LPH-AS3) could elevate the NOD2 ligands in the gut. All these results indicate that DL-endopeptidase activities of LPH and the generated NOD2 ligands are sufficient for LPH's colitis-protective effects.

## LPH regulates gut homeostasis via NOD2 signaling

We further confirmed whether NOD2 signaling mediated LPH's anti-colitis effects. We employed the *Nod2* knockout (*Nod2*$^{-/-}$) mice and found that LPH lost the ability to protect *Nod2*$^{-/-}$ mice against TNBS-induced colitis (Fig. 5a−e).

Disturbance of gut homeostasis is the essential characteristic of IBD, and NOD2 signaling is critical for establishing gut homeostasis,

**Fig. 3 | LPH is a bifunctional peptidoglycan hydrolase. a** ANTS visualization of peptidoglycan (from *M. luteus*) digests of LPH showed that it generated specific products migrating similarly to the synthetic muropeptide MDP, the red asterisk indicates the space for MDP. **b** Heatmap illustrating the peptidoglycan (from *M. luteus*) digests of different enzymes or enzyme combinations analyzed by untargeted HPLC-MS/MS. Rows indicated all detected peptidoglycan fragments; columns show different enzymes or enzyme combinations; The gray shade represents relative peptidoglycan fragments abundance as quantified by the log2-transformed area under the curve. **c** Targeted LC-MS/MS analysis of fecal MDP levels in mice gavaged with different doses (0, 1, 2.5, 5 mg/kg body weight) of LPH-containing beads for three days, *n* = 7 mice (female) per group. Data are representative of 3 independent experiments and presented as mean ± SD. Statistical analysis was performed using one-way ANOVA with Bonferroni post hoc test. **d** Representative peptidoglycan structure from gram-positive bacteria (lysine on the third amino

acid in the peptide stems), and the minimal enzymes required to generate MDP. **e** The predicted three-dimensional structure of the 3D-domain of LPH, with the electrostatic potentials and the active sites for the N-Acetyl-β-D-muramidase activity shown on the left, and the active sites for the DL-endopeptidase activity shown on the right. **f**–**g** Untargeted HPLC-MS/MS analysis of peptidoglycan (from *M. luteus*) digests of indicated enzymes or enzyme combinations. The line indicates the MDP peak. Data are representative of 3 independent experiments. **h** ANTS visualization of peptidoglycan (from *M. luteus*) digests of different enzymes or enzyme combinations, data are representative of 3 independent experiments. BSA: bovine serum albumin; PG; peptidoglycan, LPH-AS1: the D316A mutation of LPH; LPH-AS2: the D329A mutation of LPH; LPH-AS3: P295A, G297A, and T298A mutations of LPH; Mutanlolysin (M) was used as a positive control for N-Acetyl-β-D-muramidase; SagA was used as a positive control for DL-endopeptidase. Source data are provided as a Source data file.

including ameliorating inflammatory response, regulating gut microbiota, and enhancing epithelial function through protecting intestinal stem cells[25–30]. Thus, we tested LPH's ability to re-establish gut homeostasis in colitis mice. The results showed that LPH could exert comprehensively protective effects on gut homeostasis: firstly, LPH could control over-inflammation in the colon, including reducing proinflammatory cytokines TNF-α, IFN-γ, IL-6 and increasing anti-inflammatory cytokine IL-10 (Fig. 5f–i); secondly, LPH protects the intestinal barrier function, as shown reduced diffusion of FITC-dextran from intestinal mucosa to serum in Fig. 5c; thirdly, LPH could restore the dysbiotic gut microbiota induced by TNBS (Fig. 5j–l, Supplementary Data 2). All these gut homeostasis protective effects of LPH were abolished in *Nod2*−/− mice (Fig. 5a–l, Supplementary Data 2). These results reveal that LPH protects mice from colitis through NOD2-mediated gut homeostasis.

## LPH protects mice against colitis-associated CRC

Since NOD2-mediated gut homeostasis is also critical for protecting against CRC[31–33], we next evaluated LPH's protective effects on azoxymethane (AOM)/DSS induced colitis-associated colon cancer model. The model induction procedure was shown in Fig. 6a. For the mice in the treatment group, LPH was gavaged during the 3 rounds of DSS treatment periods (Fig. 6a). Loss of body weight, increased death rates, and clinical score were observed in the AOM/DSS group while treatment with LPH could mitigate these parameters significantly (Fig. 6b–d). Hematoxylin and eosin (H&E) staining revealed that AOM/DSS induced obvious hyperplasia, dysplasia, and inflammation in the colon, which was reduced by LPH treatment (Fig. 6e–h). The macroscopic tumor load parameters, including the number and size of tumor polyps, were also much lower in LPH treated group compared with the AOM/DSS group (Fig. 6i–l). All these results indicate that LPH is an enzyme of widely therapeutic potential for IBD and CRC.

## Discussion

Along with the comprehensive investigation of microbiota-host interaction, postbiotics, a new term regarding the effector of probiotics is emerging. Postbiotics are metabolites or components of the microbiota with effects on human health. Compared to probiotics, postbiotics are more stable and safer due to the clear chemical structure and no risk for infection. Thus, the identification of novel postbiotics not only provides mechanistic insights on probiotics but also is essential for developing new treatment strategies. In this study, through bioinformatics-guided screening of the secretome of clinical-established probiotics, followed by functional studies, we identified and characterized a postbiotic that is mechanistically and clinically linked with gut homeostasis. The mechanism of LPH to treat colitis is summarized in Fig. 7.

NOD2 is a classical pattern recognition receptor, activated by specific peptidoglycan fragments, such as MDP and N-acetylglucosamine-MDP[34–36]. *NOD2* polymorphisms are the strongest risk factor for Crohn's

disease (CD), one of the major types of IBD[37]. In a parallel study, we found that in addition to *NOD2* polymorphisms, deficiency in microbiota-derived NOD2 ligands also plays a critical role in CD pathogenesis[24]. We found DL-endopeptidase gene abundances decreased universally in CD patients, and supplementation of DL-endopeptidase could ameliorate colitis in mice models. The study revealed that depletion in DL-endopeptidase contributes to CD pathogenesis, and augmentation of the NOD2 pathway by enzymatic supplementation is feasible for treating colitis[24]. Here, we characterized a unique peptidoglycan hydrolase (LPH) with the following important findings: 1) LPH is shared by many clinical-established probiotics; 2) LPH which contains a 3D-domain is uncharacterized before, and has now been experimentally confirmed as a bi-functional peptidoglycan hydrolase with both N-Acetyl-β-D-muramidase and DL-endopeptidase activities that could directly generate MDP; 3) Compared to the previously well-studied NLPC-P60 domain-contained DL-endopeptidase, LPH acts on a more widely substrate due to its bi-functional peptidoglycan hydrolase. Thus, this study provides a probiotic enzyme to efficiently enhance the NOD2 signaling in vivo and reveals a common molecular mechanism of traditional probiotics.

Except for LPH, previous studies also found a secreted protein named p40 from several *Lactobacillus* strains (*L. rhamnosus* GG and *L. casei*), which could also protect mice from colitis[16]. Though the protective effects from the host perspective have been well established, including activation of epidermal growth factor receptor, regulating intestinal epithelial cell survival and growth, or promoting IgA production[38,39], the direct mechanism from the protein perspective has not been established. Other studies showed that p40 is a γ-D-Glutamyl-L-Lysyl-endopeptidase[40,41], which is critical to generating MDP, suggesting p40 may act directly through the NOD2 pathway. In addition to these important probiotic proteins, another study found that the probiotic species *Lactobacillus salivarius* also protects against colitis through the NOD2 pathway[42]. Altogether, these studies suggest that NOD2 pathway is an important anti-colitis mechanism of traditional probiotics. Further clinical trials on probiotics need to take into consideration of host genetic effects, and changes in NOD2 ligands levels should be considered when evaluating the outcomes. Additionally, the need for specific gene knock-out *Lactobacillus* strains is imperative to elucidate the primary effector(s) responsible for the observed colitis-protective effects of these probiotic strains.

NOD2 pathway is involved in a lot of gut and systemic physiological processes, including immune training, anti-inflammatory responses, epithelial regeneration and autophagy et al.[43–46]. *NOD2* mutation is also associated with many other important diseases except CD, such as Blau syndrome, asthma, arthritis and CRC[30–33,47]. Thus, the manipulation of NOD2 signaling by LPH may have broad clinical applications. Here, using AOM/DSS model, a robust and reproducible model of inflammation-associated colorectal carcinogenesis[48], we demonstrated that in addition to colitis, LPH could also alleviate colitis-associated cancer. As LPH is administrated only during the DSS

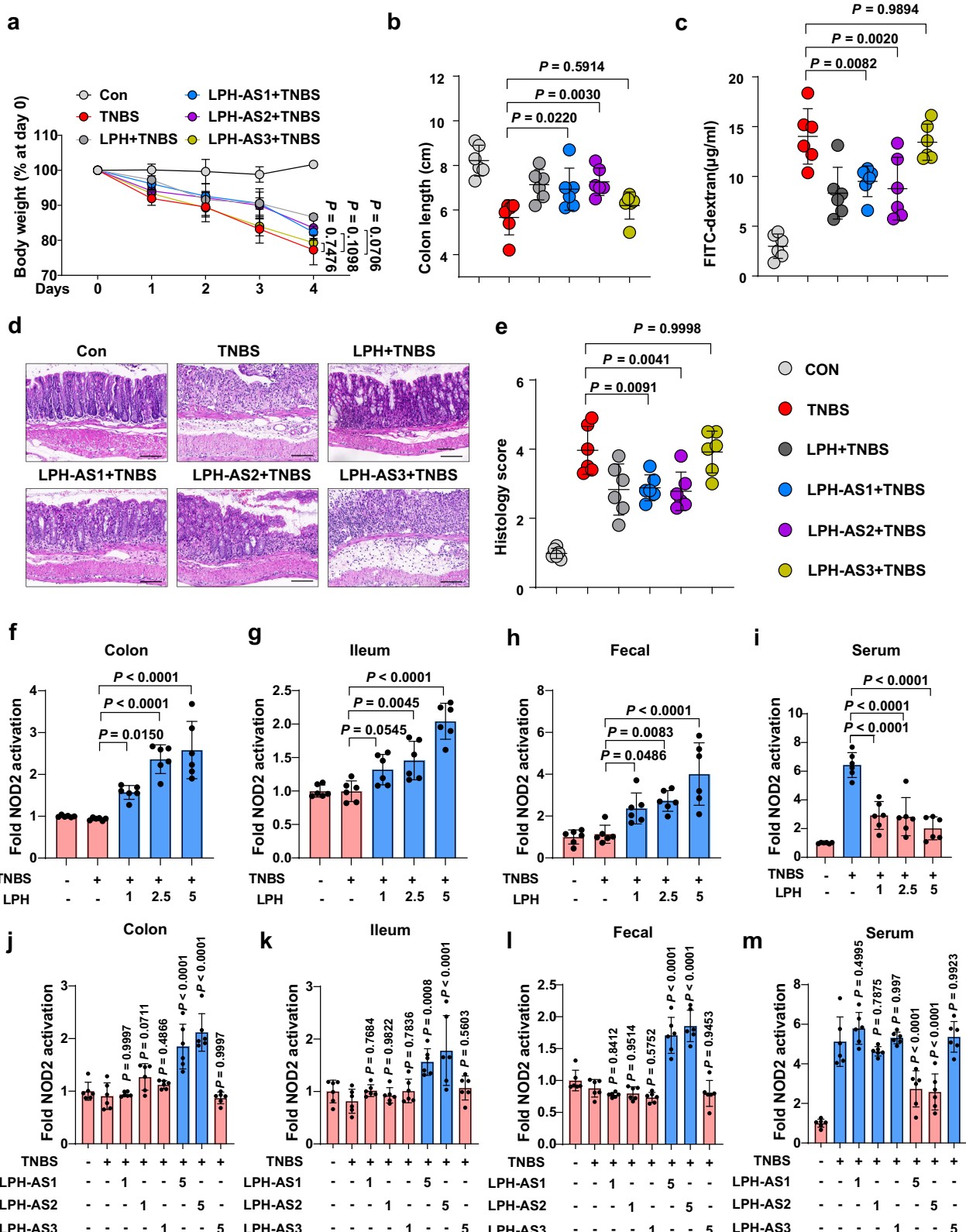

challenge, it may protect mice from CRC by dampening inflammation. Indeed, patients with long-standing IBD have an increased risk of developing CRC and anti-inflammation is beneficial in CRC in many clinical trials[49–53]. Our data suggest that LPH may have a clinical role in reducing the risk of colorectal cancer in IBD patients.

The present study has several limitations that should be noted. Firstly, the wide substrate specificity of LPH enables it to generate

NOD2 ligands from diverse bacterial sources. However, the amino acid composition of the oligopeptide chain and the cross-linked bridges within the peptidoglycan can vary significantly among different bacteria, which may lead to the production of alternative products that differ based on the peptidoglycan structure. The precise characterization of these diverse peptidoglycan products from various bacterial species was not accomplished in our study, and as such, further

**Fig. 4 | MDP mediate the colitis protective effects of LPH. a–e** The protective effects of LPH or its active site mutants (5 mg/kg body weight) on TNBS-induced colitis: body weight loss (**a**); colon length (**b**); serum FITC-dextran levels (**c**); representative H&E staining of colonic tissue, scale bar, 100 μm (**d**) and the semi-quantitative scoring of inflammation (**e**) of mice pre-treated with LPH or its different mutants. **f–i** Mice were gavaged with pectin/zein beads containing BSA (5 mg/kg body weight) or indicated dosage of LPH (mg/kg body weight), and the ability of colonic homogenate (**f**), ileal homogenate (**g**), fecal extract (**h**) or serum (**i**) to activate NOD2-expressing NF-κB reporter HEK293 cells were detected. Data are presented as fold change relative to BSA treated group (control). **j–m** Mice were gavaged with pectin/zein beads containing LPH mutants at a dosage of 1 mg/kg or 5 mg/kg body weight as indicated. The ability of colonic homogenate (**j**), ileal homogenate (**k**), fecal extract (**l**), or serum (**m**) to activate NOD2-expressing NF-κB reporter HEK293 cells were detected. Data are presented as fold change relative to BSA treated group (control). BSA: bovine serum albumin; LPH-AS1: the D316A mutation of LPH; LPH-AS2: the D329A mutation of LPH; LPH-AS3: P295A, G297A, and T298A mutations of LPH. Data are representative of 3 independent experiments and presented as mean ± SD. Each dot indicates an individual mouse ($n = 6$, female). Statistical analyses were performed using repeated measures ANOVA with Bonferroni post hoc test (**a**), one-way ANOVA with Bonferroni post hoc test (**b**, **c**, **e**, **f–m**). Source data are provided as a Source data file.

investigations are required to elucidate the potential of products beyond MDP. Another limitation of the study pertains to the lack of a crystal structure for the full-length LPH protein, which could provide additional insights into the nature of inter-domain interactions and their respective contributions to LPH's biological functions. However, the task of determining the crystal structure of a protein is a challenging and resource-intensive process that may require a considerable amount of time and effort. Despite this limitation, our study has provided significant knowledge about the functional roles of LPH, underscoring its potential as a therapeutic agent for regulating gut homeostasis.

In conclusion, this study characterizes a bacterial bi-functional PGH and reveals an important but unexplored mechanism of bacteria in maintaining gut homeostasis: secreting PGHs to manipulate murapeptides-NOD2 pathways. Taking into account the essential and broad-reaching effects of NOD2 signaling in modulating gut and systemic homeostasis, we speculated that LPH, derived from traditional probiotics with long-time-proved safety, may serve as a promising therapeutic potential in these conditions.

## Methods
### Animals
The ethics committee of Southern Medical University approved the animal protocols (No. LAEC-2022-081) and animal care and experiments were done strictly in adherence to the Southern Medical University animal care guidelines. C57BL/6 mice and New Zealand rabbits were obtained from the Animal Experimental Center of Southern Medical University. *Nod2* knockout (*Nod2*tm1cyagen, *Nod2*−/−) mice, on the background C57BL/6, were obtained from Cyagen Biosciences. *Nod2* heterozygous (*Nod2*+/−) mice were bred to generate littermate *Nod2*−/− and *Nod2*+/+ (WT) mice for experiments. All the animals were raised in specific pathogen free conditions with a strict 12 h light/12 dark shift, access to sterilized water and food ad libitum. The animal house was kept at approximately 24-26 °C and 44-46% humidity. Seven-eight weeks old mice are randomly grouped for experiments. The outcomes of mice were blindly collected. All the animals were raised in specific pathogen-free conditions with 12 h of light/12 dark shifts.

### Cell lines and culture conditions
Human NOD/NF-κB/secreted embryonic alkaline phosphatase (SEAP) reporter HEK293 cells, including HEK-Blue™ hNOD1 and HEK-Blue™ hNOD2, were obtained from InvivoGene (HEK-Blue™ NOD1, catalog number hkb-hnodl; HEK-Blue™ NOD2, catalog number hkb-hnod2). Cells were growth in Dulbecco's modified Eagle's medium (4.5 g/l glucose, GIBCO), adding 10% (v/v) fetal bovine serum (GIBCO), 2 Mm L-glutamine, 100 U/ml penicillin, 100 μg/ml streptomycin, 100 μg/ml Normocin™ (all from InvivoGene), and keeping at 37 °C in 5% CO₂, 100% humidity. Before the NOD reports experiments, cells were maintained and subcultured in a growth medium supplemented with Zeocin (100 μg/ml, InvivoGene) and Blasticidin (30 μg/ml, InvivoGene) for 3 days.

### Microbe strains
*L. paracasei* (ATCC 334), *L. casei* BL23 (ATCC 393), *L. rhamnosus* GG (ATCC 53103), *E. coli* (ATCC 25922) and *S. aureus* (ATCC 25923) were obtained from American Type Culture Collection. *L. paracasei*, *L. casei*, and *L. rhamnosus* were cultured in Man, Rogosa, and Sharpe broth at 37 °C without agitation for 24 h. *S. aureus* and *E. coli* were grown in Luria-Bertani broth at 37 °C with agitation (200 rpm) for 12-16 h.

### Secretome analysis
The 10 completed genomes from *L. casei*, *L. paracasei*, *L. reuteri*, and *L. rhamnosus* were downloaded from the NCBI genome database (Supplementary Data 1). The secretome of these strains was predicted with PSORTb (v3.0). And the proteins with extracellular scores higher than any other location scores were considered to be secreted proteins. All the secreted proteins were clustered at 75% sequence identity using CD-hit (v4.8.1).

### Bio-informatics analysis of LPH
The conserved domains of proteins were predicted using the website tool PFAM (v35.0) with default parameter or with NCBI online function. The conserved sites of protein were predicted with the ConSurf Server: https://consurf.tau.ac.il/[54]. Prediction of the three-dimensional structure was done by Modeller (v9.22-1) and visualized by PyMol (v2.4.0)[55] and AlphaFold (v2.0). Signal sequence prediction was determined using the web tool SignalP (v5.1): http://www.cbs.dtu.dk/services/SignalP/.

### Protein production and purification
A schematic of the LPH structure was showed in Fig. 2a. The Sangon Biotech (Shanghai, China) was entrusted to synthesize the following gene sequences: LPH without signal sequence (51-351 aa); D316A mutation of LPH (LPH-AS1); D329A mutation of LPH (LPH-AS2); P295A, G297A and T298A mutations of LPH (LPH-AS3); LPH-3D (51-267 aa of LPH); SagA without signal sequence (51-530 aa), LRP without signal sequence (51-357 aa) and LPP without signal sequence (51-339 aa). All these sequences contained restriction enzyme sites *EcoR* I at the 5' end and *Xho* I at the 3' end. These synthetic constructs were ligated into *EcoR* I- and *Xho* I digested pET-28a respectively, and transformed into competent *E. coli* BL21(DE3). During inducing protein expression, kanamycin was used at 50 μg/mL, and isopropyl-β-d-thiogalactoside (Sigma-Aldrich) was used at 0.5 mM for LPH, LPH-AS1, LPH-AS2, LPH-AS3, LPH-3D, LRP, and LPP, and at 1 mM for SagA. After induction of protein expression for 6 h, bacteria were pelleted and lysed ultrasonically. His-tagged proteins were purified using Histidine-Tagged Protein Purification Kit (CWBIO, China) and desalted using phosphate buffered saline (PBS) containing 20% glycerin, followed by a clearing of potential endotoxin by passing through a Detoxi-Gel Endotoxin Removing Gel (Thermo Fisher Scientific).

### Immunoblot
The anti-LPH polyclonal antibody was generated using female New Zealand white rabbits, as female is sensitive with lower doses of

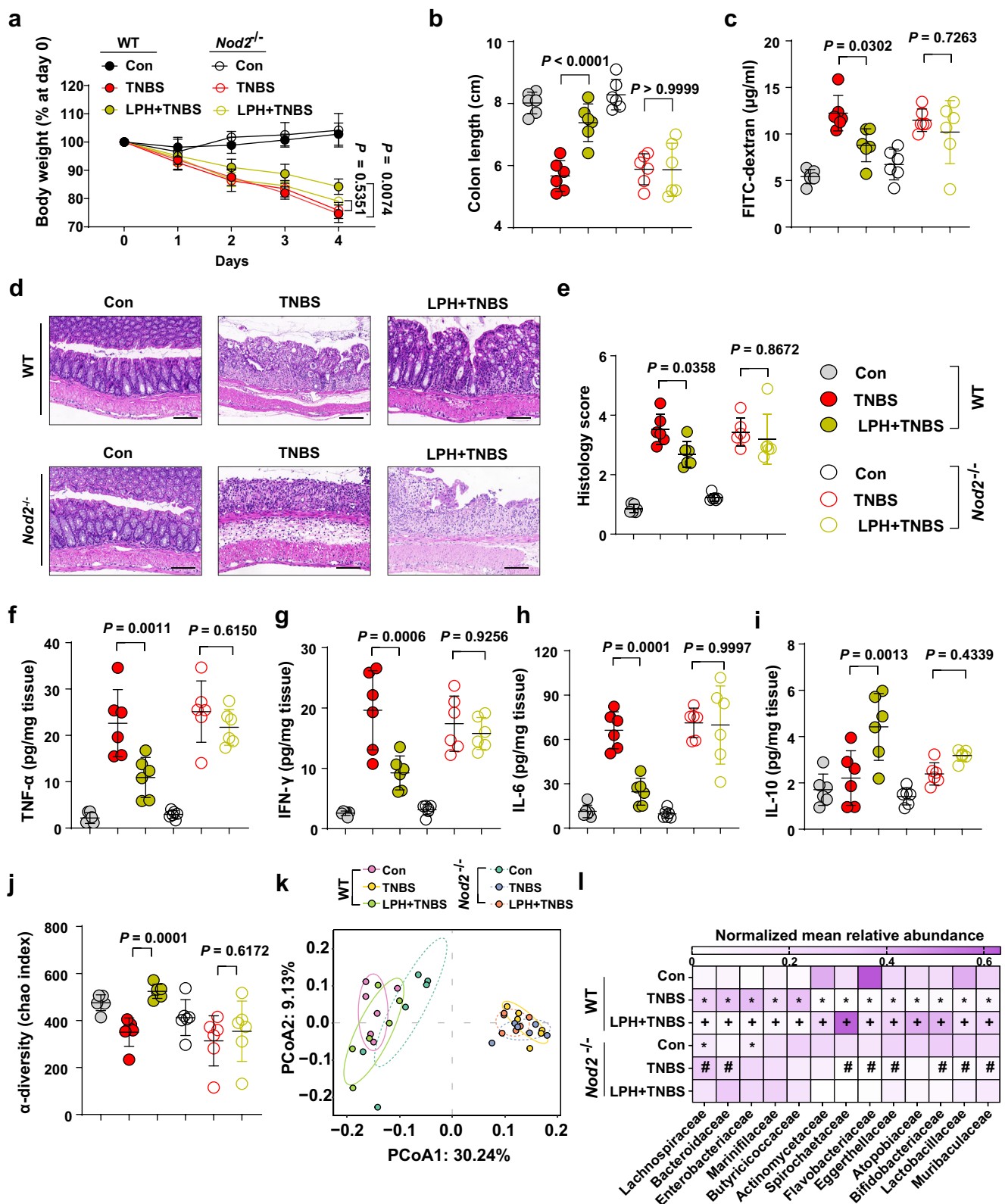

antigen and have higher response to immunization than male[56]. Female New Zealand white rabbits (6-8 weeks, 2 kg body weight) were immunized through repeated intradermal injections of LPH (1:1 emulsified in Freund's adjuvant). After the final boost, blood was collected and serum was prepared. The polyclonal antibody was purified using Pierce™ Protein A IgG Purification Kit (Thermo Fisher Scientific). Immunoblot was performed to detect the presence of LPH in the

culture supernatant of *L. casei*, *L. rhamnosus*, and *L. paracasei*. Probiotics were grown in Man, Rogosa, and Sharpe medium at 37 °C, 24 h without agitation. The cell lysates and culture supernatants were harvested, separated on SDS-polyacrylamide gel, and transferred onto polyvinylidene difluoride membranes. Membranes were blocked with 5 % skim milk and incubated with rabbit polyclonal LPH antibody (1:5000) overnight. Expression of LPH was detected using

**Fig. 5 | LPH protects mice from colitis through NOD2-mediated gut homeostasis. a–e** The disease severity of wild-type (WT) or *Nod2* knockout (*Nod2*[-/-]) mice pre-treated with BSA or LPH-containing beads (5 mg/kg body weight) for 3 days and then stimulated with TNBS for 4 days: body weight loss (**a**); colon length (**b**); serum FITC-dextran level (**c**); representative H&E staining of colonic tissue, scale bar, 100 μm (**d**) and the semiquantitative scoring of inflammation (**e**) of mice from indicated groups. **f–i** The cytokines in colon tissues were analyzed by ELISA: TNF-α (**f**), IFN-γ (**g**), IL-6 (**h**) and IL-10 (**i**). **j–l** Fecal 16 S rRNA microbial analysis: the α-diversity reflected by Chao index (**j**), the β-diversity calculated from the operational taxonomic unit (**k**), and the bacterial composition change at the family level (**l**). In

(**l**), the mean abundance of bacterial composition at the family level according to different groups, all the data are normalized according to column (L1 normalization). Only bacteria that are significantly changed by TNBS, and could be restored by LPH dependent on *Nod2* are displayed: *$p < 0.05$ compared with WT-Con, +$p < 0.05$ compared to WT-TNBS, #$p < 0.05$ compared to *Nod2*[-/-]-Con. Data are representative of 3 independent experiments and presented as mean ± SD. Each dot indicates an individual mouse (*n* = 6, female). Statistical analyses were performed using repeated measures ANOVA with Bonferroni post hoc test (**a**), two-way ANOVA with Bonferroni post hoc test (**b, c, e, f–i, j**). Source data are provided as a Source data file.

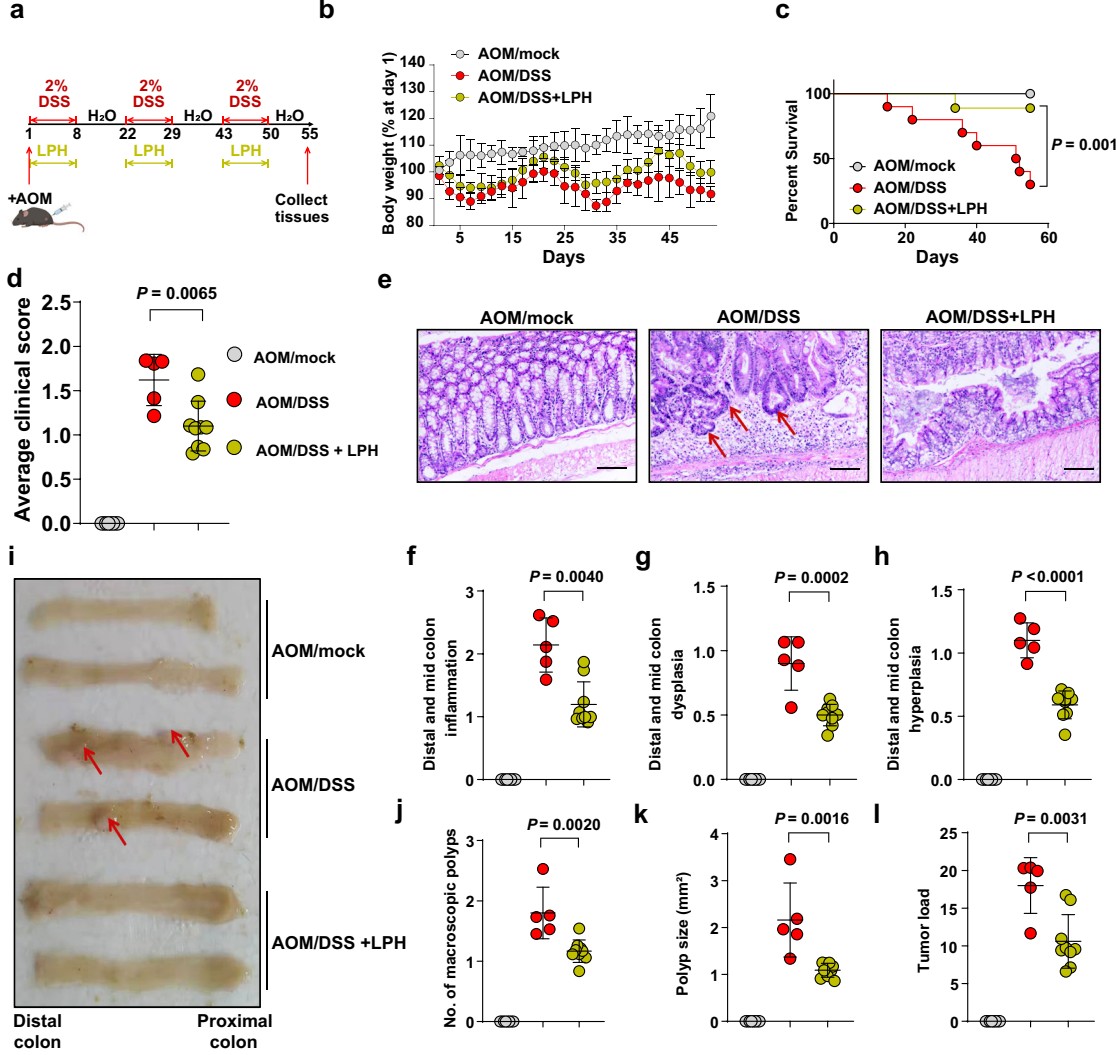

**Fig. 6 | LPH can protect mice against colitis-associated colon cancer.**
**a** Schematic of experimental design for the AOM/DSS induced colon cancer model. AOM/mock mice were intraperitoneally injected with AOM (10 mg/kg body weight) following drinking water. BSA or LPH-containing beads (5 mg/kg body weight) were orally administered during the three cycles of DSS treatment. (AOM/mock, *n* = 8; AOM/DSS *n* = 12; AOM/DSS + LPH, *n* = 10). Images were created using BioRender.com. **b, c** Loss of body weight (**b**) and survival rates (**c**) of mice treated with AOM/mock, AOM/DSS or AOM/DSS + LPH. **d** Average clinical scores of mice treat with AOM/mock (*n* = 8), AOM/DSS (*n* = 5) or AOM/DSS + LPH (*n* = 9).

**e–h** Representative H&E staining of colonic tissue, scale bar, 200 μm (**e**); semiquantitative score of colonic inflammation (**f**); dysplasia (**g**) and hyperplasia (**h**) in mice treat with AOM/mock (*n* = 8), AOM/DSS (*n* = 5) or AOM/DSS + LPH (*n* = 9). **i-l** Representative images of colons (**i**); macroscopic polyp counts (**j**); average polyp size (**k**) and tumor load (**l**) in mice treat with AOM/mock (*n* = 8), AOM/DSS (*n* = 5) or AOM/DSS + LPH (*n* = 9). AOM: azoxymethane; BSA: bovine serum albumin. Data are representative of 2 independent experiments and presented as mean ± SD. Statistical analyses were performed using Log-rank test (**c**), two-tailed unpaired Student's *t* test (**d, f, g, h, j–l**). Source data are provided as a Source data file.

goat anti-rabbit IgG antibody conjugated with horseradish peroxidase (1:5000, Proteintech, catalog number SA00001-2) and enhanced chemiluminescence reagent kit. The uncropped scans of immunoblots were provided in the Source Data file.

## Pectin/zein complex hydrogel beads
To avoid the protease attack during the oral route of administration, pectin/zein hydrogel beads were used to deliver recombinant protein to the animal colon according to previous protocols[16]. In detail, 85%

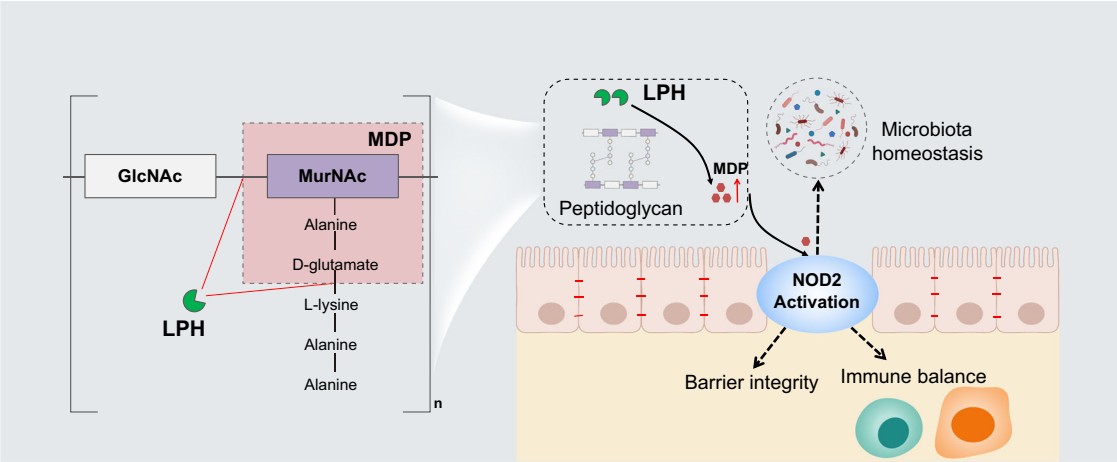

**Fig. 7 | LPH sheds NOD2 ligands to regulate gut homeostasis.** LPH is a bifunctional peptidoglycan hydrolase with both N-Acetyl-β-D-muramidase and DL-endopeptidase activities, which could directly generate MDP with high efficiency. LPH-produced MDP induces NOD2 activation to maintain gut homeostasis. Images were created using BioRender.com.

alcohol containing 0.5% (wt/vol) $CaCl_2$ was used to dissolve zein at 10 mg/ml. The recombinant protein was mixed with 60 mg/ml pectin solution to prepare the pectin/protein solution. Then, a 23 G needle was used to inhale the pectin/protein solution into a syringe and mixed with zein solution dropwise slowly. After hardening, the beads were flashed with distilled water 4–5 times. Pectin/zein beads containing bovine serum albumin (BSA) were prepared simultaneously and used as control. All beads were sized about 2 mm with 5 μg protein on average.

**Chemically induced colitis model**

The standardized procedure for chemically induced colitis models was described in ref. 57. We used female mice in all animal experiments, as they are common used in chemically induced intestinal inflammation in our lab, with robust and reproducible data[24]. To establish the TNBS-induced colitis model, female C57BL/6 mice (6–8 weeks, 18–22 g, $n = 6$ per group) were anesthetized and presensitized with 150 μl 1% TNBS (wt/vol, diluted in 4:1 mixture of acetone/olive oil, obtained from Sigma Aldrich) at the abdominal skin with a 1.5 × 1.5 cm area (Fig. 1b). Eight days later, mice were intrarectally injected with 100 μl of 2.5% TNBS (wt/vol, 1:1 in ethanol) under general anesthesia and kept vertical for 0.5 min. Control mice underwent the same procedure but were administered intrarectally with 50% ethanol. Considering that loss of the *Nod2* gene makes mice more prone to colitis, $Nod2^{-/-}$ mice (6–8 weeks, 18–22 g, $n = 6$ per group) were given a lower dosage of TNBS (1.5 %, wt/vol) than WT (littermate controls, 6–8 weeks, 18–22 g, $n = 6$ per group) to obtain a similar colitis severity, according to previous work[58]. Mice were assessed for body weight, fecal consistency, stool bleeding, and survival every day. Four days following the TNBS treatment, mice were sacrificed, the colons were separated and the lengths were recorded. A 2 cm segment of tissues from the distal colon was embedded into paraffin. After cutting into 3 μm sections, H&E staining was performed to evaluate tissue inflammation and damage. Inflammation scores were assessed as below: no signs of inflammation scored at 0–1; one to two scattered infiltrating mononuclear cells scored at 1–2; multiple foci of inflammatory cell infiltrating scored at 2–3; high level of inflammation, with increased vascular density and marked wall thickening scored at 3–4; maximal severity of inflammation, with transmural leukocyte infiltration and loss of goblet cells scored at 4–5.

For the prevention experiment, female C57BL/6 mice (6–8 weeks, 18–22 g, $n = 6$ per group) were orally administered with pectin/zein beads containing BSA (5 mg/kg body weight) or LPH (1–5 mg/kg body

weight) daily for 3 days until the TNBS challenge. For the treatment experiment, mice were orally administrated with pectin/zein beads containing BSA or LPH from one day after the TNBS challenge to the day before sacrifice. These procedures of LPH treatment were showed in Fig. 1b, i.

The procedure of oxazolone (OXA)-induced colitis is the same as that in the TNBS colitis model described in ref. 57. C57BL/6 mice (female, aged 6–8 weeks, body weight 18–22 g, $n = 6$ per group) were presensitized with 150 μl 3% OXA (wt/vol, dissolve in 4:1 mixture of acetone/olive oil, obtained from Sigma Aldrich) at the abdominal skin with a field of 1.5 × 1.5 cm. Eight days later, mice were intrarectally injected with 100 μl of 1% OXA (wt/vol, dissolve in 50% ethanol) under general anesthesia and kept vertical for 0.5 min. Control mice underwent the same procedure but were administered intrarectally with 50% ethanol. Body weight, colon lengths, and macroscopic scoring of inflammation were analyzed as described above.

The procedure for generating DSS induced colitis model was as follows: 3% DSS (Aladdin, China) solution was given to C57BL/6 mice (female, aged 6–8 weeks, body weight 18–22 g, $n = 6$ per group) in drinking water for the first 5 days, then replaced with normal drinking water until being sacrificed. Pectin/zein beads containing BSA or LPH were orally administered 3 days before DSS treatment to the day before sacrifice. Body weights were assessed daily during the experiment. At the end of the experiment, mice were sacrificed, and their colon lengths were measured. H&E staining was performed as described above. Scores of inflammation-associated histological changes were evaluated as follows: (1) Tissue damage: 0, none; 1, isolated focal epithelial damage; 2, mucosal erosions and ulcerations; 3, extensive damage deep into the bowel wall; (2) Lamina propria inflammatory cell infiltration: 0, infrequent; 1, increased neutrophils; 2, submucosal presence of inflammatory cell clusters; 3, transmural cell infiltrations. The sum of the two subscores results in a combined score ranging from 0 (no changes) to 6 (widespread cellular infiltrations and extensive tissue damage).

**Colitis-associated colon cancer model**

AOM/DSS-induced colitis-associated colorectal cancer model was generated as described in ref. 57. C57BL/6 mice (female, 6–8 weeks, 18–22 g, $n = 5$–12, as indicated in the figure legend) were given a single intraperitoneally injection of AOM solution (10 mg/kg body weight, from Sigma Aldrich), followed by 3 cycles of 2% DSS (8 days of DSS stimulation, 14 days later, Fig. 6a). LPH or BSA-contained beads were orally administered during the three cycles of DSS treatment. Weight

changes were monitored every 2 days. Clinical scores were assessed after the final round of DSS administration based on fecal consistency, rectal and fecal bleeding as according to previous protocols[59]. Mice were euthanized on day 55. Colons were separated, washed with PBS, and opened longitudinally for visual polyp counts. After H&E staining, a dissecting microscope (10× magnification) was used to identify polyps and assess the number and size of polyps (maximum dimension). The tumor loads were calculated by summing the diameters of all colon tumors per mouse.

## Intestinal permeability

The permeability of the intestinal barrier was evaluated by analyzing serum concentrations of fluorescein isothiocyanate-linked dextran (FITC-dextran, Sigma Aldrich) following oral gavage according to previous protocols[60]. Mice were gavaged with FITC-dextran (4 kDa) at a dose of 600 mg/kg body weight, and cardiac punctures were performed 4 h later. A fluorescence spectrophotometer was employed to measure the serum FITC-dextran concentration with emission wavelengths at 485 nm, and excitation wavelengths at 535 nm.

## Immunofluorescence

Fresh colon tissue was fixed in formaldehyde (4%) and processed under a standard histological process for sectioning. The LPH distribution in the colon was detected as described in ref. 16. LPH was firstly labeled with fluorescein Cy3 using Cy3® Conjugation Kit (Abcam). C57BL/6 mice (female, aged 6–8 weeks, body weight 18–22 g) were orally administrated with pectin/zein beads containing Cy3-labeled LPH or unlabeled LPH and sacrificed 6 h later. Colonic tissues were harvested and sectioned for immunofluorescence analysis. The distribution of LPH in the colon was detected using fluorescence microscopy. Fluorescence pictures were collected through NIS-Elements Basic Research Imaging Software (v3.2), and analyzed using ImageJ software (v1.49).

## ELISA

The pro-inflammatory cytokines (IFN-γ, TNF-α, IL-6), anti-inflammatory cytokine IL-10 from homogenized colon extracts were evaluated using ELISA kits (all from CUSABIO, China), according to the manufacturer's instructions.

## Fecal 16S rRNA microbial analysis

Majorbio Bio-Pharm Technology Co. Ltd. (Shanghai, China) was entrusted to perform the 16S rRNA microbial analysis of mice fecal samples. Firstly, the E.Z.N.A.® soil DNA Kit (Omega Bio-tek) was employed to isolate fecal DNA. The primers: forward, ACTCCTACGG-GAGGCAGCAG; reverse, GGACTACHVGGGTWTCTAAT, were used to amplify 16 S rRNA genes and the PCR reactions were performed in triplicate. The AxyPrep DNA Gel Extraction Kit (Axygen Biosciences) was used to purify PCR products, which were then paired-end sequenced on an Illumina MiSeq PE300 platform (Illumina, San Diego, USA) and quantified with Quantus™ Fluorometer (Promega, USA). The resulting raw data were demultiplexed and quality-filtered by Fastp (v0.20.0), and then FLASH (v1.2.7) was employed to merge those data passed quality controls. Reads were clustered into operational taxonomic units (OTUs) at the cutoff of 97% similarity using UPARSE (v7.1). Then RDP Classifier (v2.2) was used to annotate the taxonomy of the representative sequences of each OTU, using the 16S rRNA database (Silva v138) as a reference at a confidence threshold of 0.7.

## Analysis of the peptidoglycan hydrolase activity

A remazol dye-release assay was performed to measure the peptidoglycan hydrolase activity as described in ref. 61. Peptidoglycan was labeled with remazol brilliant blue as follows: peptidoglycan from *M. luteus* (1 mg in 250 μl water) was mixed with NaOH (250 μl, 400 mM) and incubated at 37 °C for 30 min; remazol brilliant blue solution (25 mM) was added and incubated at 37 °C overnight. Then 500 μl HCl (1 M) was added and mixed thoroughly, followed by centrifuging to collect peptidoglycan pellets (15,000 g for 30 min). The resulting remazol brilliant blue-covered peptidoglycan pellets were resuspended in 2 ml water. Coated peptidoglycan (100 μl) was then incubated with 20 μg LPH or its mutants in 100 μl PBS (pH 7.5) at 37 °C for 16 h. The digested products were centrifuged (20,000 g for 20 min) and the supernatants were measured at OD 595 nm.

## Peptidoglycan digestion

We digested peptidoglycan as follows: 100 μg of *S. aureus* peptidoglycan was mixed with 20 μg LPH or LPH-3D in PBS (pH 7.5) for 16 h at 37 °C. The supernatants were collected by centrifuging the mixture at 20,000 g for 10 min at 4 °C. Then, the peptidoglycan digests were filtered through 5-kDa molecular weight cut-off column, the flow-through was collected.

These peptidoglycan digests were used for evaluation of colitis protective effect in mice models and their compositions were analyzed by ANTS labeled gel electrophoresis, NOD reporter cells analysis, and HPLC-MS/MS as described below.

## ANTS labeled gel electrophoresis

Firstly, dissolve ANTS in 3:17 acetic acid/water solution to obtain 0.2 M ANTS. Then, mix 1 volume of 0.2 M ANTS with 1 volume of NaCNBH₃ (1 M, dissolve in DMSO) to obtain an ANTS reaction mixture. Peptidoglycan digests prepared as described above were dried, added with 40 μl of ANTS reaction mixture, and incubated overnight at 37 °C. ANTS labeled peptidoglycan digests were mixed 1:1 with glycerol. Labeled digests/glycerol mixture (2 μl) was loaded onto a 30% acrylamide gel without SDS. The gel was run for 8 h at 35 mA and visualized by exposure to ultraviolet light. ANTS-labeled MDP, MurNAc, GlcNAc, or MurNAc-L-Ala were run for comparison. The uncropped scans of gels were provided in the Source Data file.

## Detect the activation of NOD1 or NOD2

Human NOD/NF-κB/SEAP reporter HEK293 cells were used to monitor the activation of NOD by different samples, including in vitro peptidoglycan digests described above and serum, fecal, and tissue samples described below. As a control, NOD1/NF-κB/SEAP reporter HEK293 cells were also included. Cells were seeded in 96-well plates (25,000 cells/well) at 37 °C for 24 h before the addition of muropeptides digests, tissue extracts, serum, or fecal samples. Afterward, a diluted HEK Blue Detection media (InvivoGene) was added resulting in a volume of 200 μL. Six μg/ml iE-DAP (InvivoGene) were used as positive controls for NOD1 reporter cells and 50 ng/ml MDP (InvivoGene) were used as positive controls for NOD2 reporter cells. After incubation at 37 °C in 5% CO₂ for 12 h, SEAP activity was determined using a spectrophotometer at OD 635 nm.

## Untargeted high-performance liquid chromatography

To analyze the products of different enzyme-digested peptidoglycan, an untargeted HPLC-MS/MS analysis was performed at the Central Library of Sothern Medical University. Ten microliters of peptidoglycan digests prepared as described above were separated on a Hypersil GOLD C18 column (1.9 μm, 2.1 × 100 mm) at 50 °C. Runs were operated at 0.3 ml/min over 30 min. The mobile phases were used as follows: (A) 0.1% FA in water, (B) 0.1% FA in methanol. Products were then analyzed with Thermo Scientific™ Orbitrap Fusion™ Tribrid™ using the positive mode of electrospray ionization, and mass range from 50 to 2000 m/z was acquired.

## Growth curve

To determine if LPH can lyse *E. coli* or *S. aureus*, the 50 μl overnight cultures were added into 5 ml fresh Luria-Bertani broth. LPH was

supplemented at a final concentration of 20 μg/ml. The new inoculum was incubated at 37 °C with agitation. OD600 was measured at 0.5 h intervals.

## Fecal, serum, and tissue sample preparation

After being weighed by an analytical balance (sample weights 150–200 mg), 500 μl/100 mg high-grade water (18 MOmwas) was added to each sample and then mixed by vortexing at max speed for 3–5 s. A sonication bath was carried out at 25 °C for 10 min. The supernatant was obtained by centrifuging the mixture at 20,000 g for 10 min. The serum samples were centrifuged to collect the supernatants (20,000 $g$ for 10 min). The colonic tissues were homogenated with PBS (250 μl PBS per 100 mg of tissue) and the supernatants were cleared by centrifugation using 20,000 $g$ for 10 min. All the resulting fecal, serum, and tissue supernatants were filtered through 0.22 μm pore size filters. The flow-through was placed at −80 °C for one night before examining for MDP levels utilizing NOD2/NF-κB/SEAP reporter HEK293 cells described above or targeted LC-MS/MS methods described below.

## MDP quantification with LC-MS/MS

All fecal samples from LPH-treated mice were analyzed for MDP levels by a targeted metabolomic method. The fecal samples were prepared as described above. A quality control sample was prepared by mixing with 5 μl of each sample. Standard curves were prepared in high-grade water in a concentration from 12.8 ng/ml to 1600 ng/ml. One hundred microliter samples were separated on a Prelude SPLC System using Waters Acquity UPLC HSS T3 column (2.1 × 100 mm, 1.8 μm) at 40 °C. The mobile phases used as follows: (A) 0.1% FA in water, (B) 0.1% FA in acetonitrile, with gradient of 0 min (2% B), 0.5 min (30% B), 3.5 min (85% B), 5 min (85% B), 6 min (2% B) at flow rate 0.25 ml/min.

Multiple reaction monitoring scanning modes were used for MS analysis (TSQ Quantiva triple quadrupole mass spectrometer). An H-ESI source in positive mode was used. The parameters were as follows: the collision energy at 10.25 V for the ion pair (493.213 $m/z$, 475.222 $m/z$) and 16.62 V for the ion pair (493.213 $m/z$, 329.183 $m/z$). Peak detection was visualized with the Thermo Xcalibur software (v4.0).

## Quantification and statistical analysis

All the analyses in this study were done by GraphPad Prism (v8.0.2). Data are representative of 2–3 independent experiments and presented as mean ± standard deviation (SD). Body weight changes of mice were analyzed by repeated measures ANOVA, followed by the Bonferroni post hoc test. Differences between WT and $Nod2^{-/-}$ mice were analyzed using two-way ANOVA and Bonferroni post hoc test. Survival rate comparisons were analyzed with the log-rank test. Other datasets were analyzed using the one-way ANOVA followed by the Bonferroni post hoc test. Two side $p$ value less than 0.05 was considered significant. The number of repeats for each experiment and detailed descriptions of statistical tests are specified in the results section and the respective figure legends.

## Reporting summary

Further information on research design is available in the Nature Portfolio Reporting Summary linked to this article.

## Data availability

The authors declare that all data supporting the findings of this study are available within this paper and its Supplementary Files. The Microbiome (16S rRNA sequencing) data generated in this study have been deposited into the Sequence Read Archive (SRA) in National Center for Biotechnology Information (NCBI) under the accession NO. PRJNA973702. The Silva database used in 16S rRNA microbial analysis is download from https://www.arb-silva.de/. The genomes of probiotics are downloaded from NCBI: https://www.ncbi.nlm.nih.gov/genome/. Source data are provided with this paper.

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

## Acknowledgements

The authors thank Dr. Sheng-He Huang for kindly providing *L. rhamnosus* GG strains. This work was supported by the following fundings: National Key Research and Development Program of China (2022YFA0806400 to Z.H.); National Natural Science Foundation of China (81925026 to Z.H.; 82130068 to Z.H.; 31900101 to G.J.; 82272387 to G.J.; 32170111 to H.X.); Natural Science Foundation of Guangdong Province of China (2021A1515010429 to G.J.; 2021A1515010455 to H.X.; 2022A1515010402 to H.X.); Guangzhou Key Research Program on Brain Science (202206060001 to Z.H.).

## Author contributions

G.J., H.X., S.K., X.Q., W.L., and Y.P. performed most of the bioinformatics analysis, and animal experiments, and designed and

wrote the manuscript. J.J., Z.N., and L.B. handled the fecal sample collection, preparation, and targeted metabolic analysis. Z.H. preconceived, designed, and supervised all studies, and the drafting and editing of the manuscript. All authors contributed to the critical review of the manuscript.

## Competing interests

The authors declare no competing interests.
