## [Peer Review File · Nature Communications]

A probiotic bi-functional peptidoglycan hydrolase sheds NOD2 ligands to regulate gut homeostasis in female miceREVIEWER COMMENTS

Reviewer #1 (Remarks to the Author):

The study by Xiaolong He *et al.* investigates the effect of an enzyme released from probiotic *Lactobacillus* strains which is shown to mediate the release of MDP from bacterial peptidoglycan. First the authors demonstrate the effect of oral LPH in a TNBS model, where they show a protection, both in pre-treatment as well as therapeutic setting. They analyze the domain structure of the enzyme and show that the 3D domain is required for peptidoglycan degradation and the therapeutic effect in the TNBS model. They next use biochemical methods to demonstrate that indeed LPH is a bifunctional enzyme with combined muramidase and DL endopeptidase function. In mutation experiments they show that the endopeptidase activity is required for effective protection against colitis. In several *in vivo* settings employing NOD2 KO mice they show that the amelioration of TNBS colitis by LPH is mediated via this pathway. Lastly, they claim an anti-tumorigenic function of LPH in an AOM/DSS model. While the findings present a massive body of data of the effect of LPH, I have several concerns which hamper the significance of the results.

1. In a previous paper, the authors (Gao *et al.*, CHM) have more or less shown the same effect using bacterial DL endopeptidases in a DSS colitis model (plus characterization of CD patients). This study clearly decreases the novelty of the findings, as one would expect exactly this effect from an enzyme with DL endopeptidase activity.
2. The microbiological /community composition analyses are not convincing, at some point the authors state that LPH is not interfering with growth of bacteria, but wouldn't lysis of bacterial cell walls be expected to do exactly this? Also the beta diversity analysis shows a 30% explanation in axis 1 between WT and NOD2, which is not really influenced by treatment (but the authors claim correction of dysbiosis), the panel M then shows a bubble plot of bacterial "family level" analyses, where however no baseline differences are visible, which is counterintuitive when looking at beta-diversity PCA. As the authors used littermates for experiments, the seemingly huge effect at baseline between genotypes is surprising and left unexplained. This hampers the interpretation of all animal-based experiments.
3. In the AOM experiment there is a significant loss of animals early on, when tumors are usually not present, which could explain the death of the animal. Also the LPH is given only during the DSS phases, so the best guess is that LPH dampens inflammation (as in the CHM paper), which is required for tumor formation. The early loss of animals in the AOM group without LPH is likely to be explained by this effect and is not surprising.
4. The beta defensin histologies are not convincing. A major claim for NOD2 has been that it changes alpha defensin in CD. The observed changes of beta-defensin represent the presence or absence of a functioning colonic epithelial layer and not a direct effect of LPH. What about alpha defensin in the (unaffected?) ileum?
5. How does the study link to an ongoing clinical trial with the exact same official title (<https://clinicaltrials.gov/ct2/show/NCT04924686>)?

Reviewer #2 (Remarks to the Author):

In this manuscript, Gao et al. examined compounds secreted by probiotic strains currently used in clinical trials. The authors identified a common secreted compound called LPH which acted as a bi-functional peptidoglycan hydrolase. In this capacity, LPH generated muramyl dipeptide which activated NOD2 and limited colitis and colitis-induced cancer. Overall, the manuscript is well written, the figures are excellent and the discussion is appropriate. A major concern is that the statistics are not properly executed. Additionally, a vehicle delivery control (pectin) should be included alone to confirm that this compound does not affect inflammation independent of LPH. Finally, some of the staining needs to be redone. However, with these changes and the other comments noted below, this work could provide the first description of the anti-colitis properties of LPH and expand our knowledge of probiotic secreted anti-inflammatory compounds.

Comments:

1. Lactobacillus species were recently reclassified. Please update your nomenclatures as follows:
Limosilactobacillus reuteri
Lacticaseibacillus casei
Lacticaseibacillus paracasei
Lacticaseibacillus rhamnosus
2. Does Figure 1D depict SEM or Stdev? I would expect far more variation in the TNBS colitis in terms of weight. Additionally, the word TNBS in the figure legend of Figure 1D is cut off.
3. Does the pectin delivery system affect the colitis at all? The colitis models should be repeated with the pectin delivery system (without the LPH) to confirm that the delivery vehicle is not influence the inflammation. These are important controls that should be included.
4. The X axis is missing from several graphs in Figure 1 and 2 (for example Figure 1E, F, H, J, L....). Please modify the graphs with the titles underneath the x-axis.
5. The staining for B-defensin-2 appears to be non-specific. I recommend repeating the staining with immunostaining (fluorescence) and providing an inset with a close-up view of the tissue. Please include scale bars in the images as well.
6. Relative optical density does not make sense for a y-axis in Figure 1J. Since the data is quantified from the images, it y-axis should say something like "relative staining intensity".
7. Based on the histology, weight and colon length, I would not say that LPH ameliorated the colitis. The histology in particular indicates that there is still colitis in LPH treated mice, just not to the same extent as the untreated mice. It would be safer to say that LPH moderately reduced the colitis.
8. Since only 2 doses were selected, it is inappropriate to say that there was a dose dependent response in line 103. I recommend removing this statement.
9. The Occludin staining in Figure 5J is non-specific. I recommend doing immunostaining instead and providing an inset with a close-up view of the tissue. And rename the y-axis of the Figure 5K.
10. The body weight changes need to be analyzed by a repeated measures ANOVA (since they are taken over time) – not a 2-Way ANOVA. Please rerun the statistics for these types of data.
11. Additionally, all the WT and NOD2KO mouse data needs to be analyzed by a 2-WAY ANOVA rather

than a One Way ANOVA.

Reviewer #3 (Remarks to the Author):

A few queries need to be addressed:

1. How did the authors derive to the statement line no. 86 and 87". The representative sequence .. of this cluster...

2. How did the author obtain LPH? chromatography.. the details missing

3. Data of only 3D domain and peptidoglycan hydrolytic activity is missing.

4. LPH shows wide substrate specificity. The products formed may differ .. therefore, author needs to discuss in detail. Domain interaction study

5. Secondary str prediction to the level of domain is essential

6. Individual domains of LPH cloning, and functions and their cooperative functions needs to be checked. Whether there is a inter domain interaction?

7. Which domain of LPH protects Mice from colitis is missing

8. The author would have considered Lactobacillus as a control for LPH

9. What is the amount of production of secretion of LPH by Lactobacillus spp how do they differ?

10. Commercial MDP should have been the best control.

LPH does it maintain Eubiosis. Does it overcome the problem of Dysbiosis. Microbiota data is essential

11. Lane no- 191-192

Both activities are needed for LPH protection effect. But DL-endopetidase function is more... required why? rationale

12. P40 and LPH both functions on NOD2 pathway. Are they not competetors?

13. Rationale is required for Fig.O

14. Why is the Bifido content gets reduced after treatment with LPH and maintaining enterobacteriace constant.

The above reasons may make the MS effective therefore, they may be considered to look in very seriously.

Point by point responses

Reviewer #1 (Remarks to the Author):

The study by Xiaolong He cum suis investigates the effect of an enzyme released from probiotic Lactobacillus strains which is shown to mediate the release of MDP from bacterial peptidoglycan. First the authors demonstrate the effect of oral LPH in a TNBS model, where they show a protection, both in pre-treatment as well as therapeutic setting. They analyze the domain structure of the enzyme and show that the 3D domain is required for peptidoglycan degradation and the therapeutic effect in the TNBS model. They next use biochemical methods to demonstrate that indeed LPH is a bifunctional enzyme with combined muramidase and DL endopeptidase function. In mutation experiments they show that the endopeptidase activity is required for effective protection against colitis. In several in vivo settings employing NOD2 KO mice they show that the amelioration of TNBS colitis by LPH is mediated via this pathway. Lastly, they claim an anti-tumorigenic function of LPH in an AOM/DSS model.

While the findings present a massive body of data of the effect of LPH, I have several concerns which hamper the significance of the results.

Response to general comment: We sincerely thank the reviewer for the positive remarks along with the many helpful suggestions to improve the manuscript. We have attempted to address the reviewer's concerns by performing additional experiments, analyses and revisions. Below please find a point-by-point response to the comments.

Comment 1: In a previous paper, the authors (Gao et al., CHM) have more or less shown the same effect using bacterial DL endopeptidases in a DSS colitis model (plus characterization of CD patients). This study clearly decreases the novelty of the findings, as one would expect exactly this effect from an enzyme with DL endopeptidase activity.

Reply 1: We thank the reviewer for this insightful comment. As you see, our previous work has clearly demonstrated that DL-endopeptidase, a peptidoglycan hydrolase that digesting D-glutamic acid-meso-A₂pm linkage, is critical for NOD2 ligands generation. The DL-endopeptidase gene abundances decreased universally in patients with Crohn's disease (CD), and supplementation of DL-endopeptidase could ameliorate colitis in mice models. The study revealed that depletion in DL-

endopeptidase contributes to CD pathogenesis, and augmentation of the NOD2 pathway by enzymatic supplementation is feasible for treating colitis. Here, we characterized a new peptidoglycan hydrolase (LPH) with the following novel and important findings:

- 1) LPH is shared by many clinical-established probiotics such as *Lactobacillus casei* (*L. casei*), *Lactobacillus paracasei* (*L. paracasei*) and *Lactobacillus rhamnosus* (*L. rhamnosus*).
- 2) LPH which contains a 3D-domain is uncharacterized before, and was first proved to be a bifunctional peptidoglycan hydrolase with both N-Acetyl- β -D-muramidase and DL-endopeptidase activities that could directly generate muramyl dipeptide (MDP) without mutanolysin.
- 3) Compared to the NLPC-P60 domain-contained DL-endopeptidase, we found LPH with 3D-domain act on a more widely substrate due to its bifunctional peptidoglycan hydrolase.
- 4) We first experimentally proved that LPH could also ameliorate colitis-associated colon cancer.

Thus, this study provides a new probiotic enzyme to efficiently enhance the NOD2 signaling *in vivo* and reveals a common molecular mechanism of traditional probiotics. Taken that NOD2 signaling is implicated in a wide range of physiological activities, including immune training, epithelial regeneration, insulin resistance, vaccination response and promoting the effects of anti-PD-L1 immunotherapy¹⁻⁵, supplementation of LPH may have substantial biological and clinical importance.

We have added these contents in the Discussion Section of revised manuscript as follows:

Discussion (Line 252-267): *“In a parallel study, we found that in addition to NOD2 polymorphisms, deficiency in microbiota-derived NOD2 ligands also plays a critical role in CD pathogenesis³⁷. We found DL-endopeptidase gene abundances decreased universally in CD patients, and supplementation of DL-endopeptidase could ameliorate colitis in mice models. The study revealed that depletion in DL-endopeptidase contributes to CD pathogenesis, and augmentation of the NOD2 pathway by enzymatic supplementation is feasible for treating colitis³⁷. Here, we characterized a unique peptidoglycan hydrolase (LPH) with the following novel and important findings: 1) LPH is shared by many clinical-established probiotics; 2) LPH which contains a 3D-domain is uncharacterized before, and was first proved to be a*

bi-functional peptidoglycan hydrolase with both N-Acetyl-β-D-muramidase and DL-endopeptidase activities that could directly generate MDP; 3) Compared to the previously well-studied NLPC-P60 domain-contained DL-endopeptidase, LPH acts on a more widely substrate due to its bi-functional peptidoglycan hydrolase. Thus, this study provides a new probiotic enzyme to efficiently enhance the NOD2 signaling in vivo and reveals a common molecular mechanism of traditional probiotics.”

Discussion (Line 286-295): *“Thus, the manipulation of NOD2 signaling by LPH may have broad clinical applications. Here, using AOM/DSS model, a robust and reproducible model of inflammation-associated colorectal carcinogenesis⁴⁸, we demonstrated that in addition to colitis, LPH could also alleviate colitis-associated cancer. As LPH is administrated only during the DSS challenge, it may protect mice from CRC by dampening inflammation. Indeed, patients with long-standing IBD have an increased risk of developing CRC and anti-inflammation is beneficial in CRC in many clinical trials ⁴⁹⁻⁵³. Our data suggest that LPH may have a clinical role in reducing the risk of colorectal cancer in IBD patients.”*

References:

1. Cavallari, J. F. et al. Muramyl Dipeptide-Based Postbiotics Mitigate Obesity-Induced Insulin Resistance via IRF4. *Cell Metab.* **25**, 1063-1074 (2017).
2. Griffin, M. E. et al. Enterococcus peptidoglycan remodeling promotes checkpoint inhibitor cancer immunotherapy. *Science* **373**, 1040-1046 (2021).
3. Kim, D. et al. Nod2-mediated recognition of the microbiota is critical for mucosal adjuvant activity of cholera toxin. *Nat. Med.* **22**, 524-530 (2016).
4. Mulder, W. et al. Therapeutic targeting of trained immunity. *Nat. Rev. Drug. Discov.* **18**, 553-566 (2019).
5. Nigro G, Rossi R, Commere PH, Jay P, Sansonetti PJ. The Cytosolic Bacterial Peptidoglycan Sensor Nod2 Affords Stem Cell Protection and Links Microbes to Gut Epithelial Regeneration. *Cell Host Microbe* **15**, 792-798(2014).

Comments 2: The microbiological/community composition analyses are not convincing, at some point the authors state that LPH is not interfering with growth of bacteria, but wouldn't lysis of bacterial cell walls be expected to do exactly this? Also the beta diversity analysis show a 30% explanation in axis 1 between WT and NOD2, which is not really influenced by treatment (but the authors claim correction of dysbiosis), the panel M then show a bubble plot of bacterial “family level” analyses,

where however no baseline differences are visible, which is counterintuitive when looking at beta-diversity PCA. As the authors used littermates for experiments, the seemingly huge effect at baseline between genotypes is surprising and left unexplained. This hampers the interpretation of all animal-based experiments.

Reply 2: We thank the reviewer for the comments with respect to the direct influence of LPH on bacterial growth, as well as the influence of LPH on gut microbiota composition in the context of colitis. Our *in vitro* experiment shown LPH could lyse peptidoglycan, and generate NOD2 ligands from commercial peptidoglycan or cultured bacteria. The survival curve showed LPH didn't influence bacteria growth (this experiment was repeated for 3 times). These results were in accordance with the previous study, which found SagA, a NLPC-P60 DL-endopeptidase could also lyase bacterial cell walls but didn't interfere with the growth of bacteria (The statement in the article: "In culture, recombinant SagA had no effect on *E. coli* growth rate")¹. Taken that the peptidoglycan is constantly released from bacteria during cell growth and division²⁻³, this phenomenon may be explained by the possibility that recombinant LPH generates NOD2 ligands from the released peptidoglycan from bacteria, instead of lyase the peptidoglycan directly from the cell wall of live bacteria, which may have some protective mechanism from being lysed by foreign hydrolase.

For the beta diversity analysis of the gut microbiota composition, the reviewer may have misread the experimental grouping as shown in **Original Figure 5N (also Author Response Figure 1a below)**. In WT mice, TNBS obviously influences the microbiota compositions (Control in purple, TNBS in yellow), which is almost restored by LPH treatment on axis one (LPH treatment in green). In *Nod2*^{-/-} mice, TNBS obviously influence the microbiota compositions (Control in dark green, TNBS in blue), while LPH couldn't restore the microbiota (LPH treatment in orange). These results indicate that LPH regulates gut microbiota dysbiosis through the NOD2 pathway. At baseline, there is some difference between WT (purple) and *Nod2*^{-/-} littermates (dark green) as shown by beta-diversity PCoA, but this difference is much smaller than the TNBS treatment (**Author Response Figure 1a**). This minor difference at baseline is acceptable since NOD2 deficiency has been reported to cause dysbiosis in mice⁴⁻⁶. Moreover, to make it more clear, we provided the bar plot of values on axis one according to the different groups below (**Author Response Figure 1b**), along with the original data on the OTU level of all individual mice (**Supplementary Table 2** in the revised manuscript). Furthermore, we apologize for

the color of the line we used in the PCoA plot, which may not clear enough. Thus, we retyped the line of *Nod2*^{-/-} mice as dotted below (**Author Response Figure 1c, Revised Manuscript Figure 5k**).

Author Response Figure 1: Microbiota 16S rRNA analysis of WT/*Nod2*^{-/-} mice. (a) Figure 5O in the original manuscript. (b) the bar plot of values on axis one of the beta-diversity PCoA plot. (c) Replotted PCoA in the revised manuscript.

References:

1. Kavita, J. et al. A secreted bacterial peptidoglycan hydrolase enhances tolerance to enteric pathogens, *Science* **23**,1434-1437(2016).
2. Mamou, G. et al. Peptidoglycan maturation controls outer membrane protein assembly. *Nature* **606**,953-959(2022).
3. Egan, A. J. F., Errington, J. & Vollmer, W. Regulation of peptidoglycan synthesis and remodelling. *Nat Rev Microbiol.* **18**,446-460(2020).
4. Couturier-Maillard A. et al. NOD2-mediated dysbiosis predisposes mice to transmissible colitis and colorectal cancer. *J. Clin. Invest.* **123**,700-711(2013).
5. Chu, H. et al. Gene-microbiota interactions contribute to the pathogenesis of inflammatory bowel disease. *Science* **352**,1116-20(2016).
6. Rochereau, N. et al. NOD2 deficiency increases retrograde transport of secretory IgA complexes in Crohn's disease. *Nat. Commun.* **12**,261(2021).

Comment 3. In the AOM experiment there is a significant loss of animals early on, when tumors are usually not present, which could explain the death of the animal. Also the LPH is given only during the DSS phases, so the best guess is that LPH dampens inflammation (as in the CHM paper), which is required for tumor formation. The early loss of animals in the AOM group without LPH is likely to be explained by this effect and is not surprising.

Reply 3: We appreciate this insightful comment on the possible mechanism of LPH's protective effects on colorectal cancer, and totally agree that LPH may ameliorate tumors by dampening gut inflammation. The AOM-DSS model is an efficient animal model which is robust and reproducible, and it has emerged to become one of the most frequently used models to study inflammation-associated colorectal carcinogenesis¹. Loss of animals (C57/BL6) was observed typically after the first round of 2% DSS treatment in the dose of AOM (10 mg/kg body weight) as described in many investigations²⁻⁶. Actually, patients with long-standing inflammatory bowel disease (IBD) have an increased risk of developing colorectal cancer⁷⁻⁹ and anti-inflammation is beneficial in colorectal cancer in many clinical trials¹⁰⁻¹¹. Together, these results suggest that LPH may have a clinical role in reducing the risk of colorectal cancer in IBD patients. We have added these contents in the Discussion Section of the revised the manuscript, as follows:

Discussion (Line 286-295): *“Thus, the manipulation of NOD2 signaling by LPH may have broad clinical applications. Here, using AOM/DSS model, a robust and reproducible model of inflammation-associated colorectal carcinogenesis⁴⁸, we demonstrated that in addition to colitis, LPH could also alleviate colitis-associated cancer. As LPH is administrated only during the DSS challenge, it may protect mice from CRC by dampening inflammation. Indeed, patients with long-standing IBD have an increased risk of developing CRC and anti-inflammation is beneficial in CRC in many clinical trials⁴⁹⁻⁵³. Our data suggest that LPH may have a clinical role in reducing the risk of colorectal cancer in IBD patients.”*

References:

1. Tanaka, T., Kohno, H., Suzuki, R., Yamada, Y., Sugie, S. & Mori, H. A novel inflammation-related mouse colon carcinogenesis model induced by azoxymethane and dextran sodium sulfate. *Cancer Sci.* **94**, 965-973(2003).
2. Wilson, J.E. et al. Inflammasome-independent role of AIM2 in suppressing colon tumorigenesis via DNA-PK and Akt. *Nat. Med.* **21**, 906-913(2015).
3. Zheng, H., Lu, Z., Wang, R., Chen, N. & Zheng, P. Establishing the colitis-associated cancer progression mouse models. *Int. J. Immunopathol. Pharmacol.* **29**,759-763(2016).
4. Arnesen, H. et al. Induction of colorectal carcinogenesis in the C57BL/6J and A/J mouse strains with a reduced DSS dose in the AOM/DSS model. *Lab Anim. Res.* **37**, 19 (2021).

5. Yang, H. et al. Triclocarban exposure exaggerates colitis and colon tumorigenesis: roles of gut microbiota involved. *Gut Microbes* **12**, 1690364 (2020).
6. Singh, K. et al. Ornithine Decarboxylase in Macrophages Exacerbates Colitis and Promotes Colitis-Associated Colon Carcinogenesis by Impairing M1 Immune Responses. *Cancer Res.* **78**, 4303-4315 (2018).
7. Schmitt, M. & Greten, F. R. The inflammatory pathogenesis of colorectal cancer. *Nat. Rev. Immunol.* **21**, 653-667 (2021).
8. Tabung, F. K., et al. Association of Dietary Inflammatory Potential With Colorectal Cancer Risk in Men and Women. *JAMA Oncol.* **4**, 366-373 (2018).
9. Rhodes, J. M., Campbell B J. Inflammation and colorectal cancer: IBD-associated and sporadic cancer compared. *Trends Mol. Med.* **8**, 10-16 (2002).
10. Wang, D. & DuBois, R. N. The role of anti-inflammatory drugs in colorectal cancer. *Annu. Rev. Med.* **64**, 131-144 (2013).
11. Sandler, R. S. et al. A randomized trial of aspirin to prevent colorectal adenomas in patients with previous colorectal cancer. *N. Engl. J. Med.* **348**, 883-890 (2003).

Comment 4. The beta defensin histologies are not convincing. A major claim for NOD2 has been that it changes alpha defensin in CD. The observed changes in beta-defensin represent the presence or absence of functioning colonic epithelial layer and not a direct effect of LPH. What about alpha defensin in the (unaffected?) ileum?

Reply 4: Thanks for this important suggestion. We completely agree that changes of beta-defensin represent the presence or absence of a functioning colonic epithelial layer, and not a direct effect of LPH, as the colonic epithelial layer, which is the main source of defensins¹, is damaged by TNBS and restored by LPH treatment. Thus, we remove these results of defensin in the manuscript accordingly. Indeed, some previous study reported that *NOD2* mutations in CD are associated with diminished mucosal α -defensin expression (mainly DEFA5 and DEFA6)²⁻³, we thus detect the expression of Defa5 and Defa6, also the mainly expressed α -defensins (Defa3 and Defa20) in the **unaffected intestinal mucosa of mice**. As shown in **Author Response Figure 2**, treatment of LPH has no effect on the transcriptional expression of α -defensin in the **unaffected intestinal mucosa of mice**. This result is not surprising, as there are many controversial findings about the role of NOD2 in α -defensin expression. Simms et al. reported that decreased α -defensin expression is a consequence of inflammation per se and unrelated to the *NOD2* polymorphic state, implying that altered antimicrobial

function might be secondary to active CD rather than a contributor to intestinal inflammation⁴. Also, Shanahan et al. found that Nod2 does not directly regulate Paneth cell antimicrobial activity in mice⁵. The biochemical basis for this dichotomous property of Nod2 function remains unknown. Accordingly, we revised the statement about the α -defensins in the revised manuscript, as follows:

Results (Line 214-217): “secondly, LPH protects the intestinal barrier function, as shown reduced diffusion of FITC-dextran from intestinal mucosa to serum in **Fig. 5c**; thirdly, LPH could restore the dysbiotic gut microbiota induced by TNBS (**Fig. 5j-5l, Supplementary Table 2**).”

Author Response Figure 2. Transcriptional expression of alpha-defensins in ileal tissue of mice. C57BL/6 mice were gavaged with pectin/zein beads containing BSA or LPH (5 mg/kg body weight) for three days. Ileal tissues were harvested for qPCR analysis. **a-d** Transcriptional expression of *Defa3* (**a**), *Defa5* (**b**), *Defa6* (**c**) and *Defa20* (**d**) in BSA or LPH treated mice. Each dot indicates an individual mouse (n = 6, female). Data are representative of 2 independent experiments and presented as mean \pm SD. ns: not significant, analyzed by Student T-test.

References:

1. Ayabe, T. et al. Secretion of microbicidal alpha-defensins by intestinal Paneth cells in response to bacteria. *Nat. Immunol.* **1**, 113-118 (2000).
2. Wehkamp, J. et al. NOD2 (CARD15) mutations in Crohn's disease are associated with diminished mucosal alpha-defensin expression. *Gut* **53**, 1658-64 (2004).
3. Bevins, C. L., Stange, E. F. & Wehkamp, J. Decreased Paneth cell defensin expression in ileal Crohn's disease is independent of inflammation, but linked to the NOD2 1007fs genotype. *Gut* **58**, 882-883 (2009).

4. Simms, L. A., Doecke, J. D., Walsh, M. D., Huang, N., Fowler, E. V. & Radford-Smith, G. L. Reduced alpha-defensin expression is associated with inflammation and not NOD2 mutation status in ileal Crohn's disease. *Gut* **57**, 903-910 (2008).
5. Shanahan, M. T. et al. Mouse Paneth cell antimicrobial function is independent of Nod2. *Gut* **63**, 903-910 (2014).

Comment 5. How does the study link to an ongoing clinical trial with the exact same official title (<https://clinicaltrials.gov/ct2/show/NCT04924686>)?

Reply 5: The project NCT04924686 is an ongoing clinical trial initiated by our team (2020.05 to 2023-08) plan to systemically profiled peptidoglycan fragments in a diversity of diseases (inflammatory bowel disease; colorectal cancer; type 2 diabetes; atherosclerotic cardiovascular disease). This project is inspired by series studies about the role of NOD2 in maintaining gut homeostasis and different diseases¹⁻⁷, which including the CHM article mentioned above, this work, and some unpublished data.

References:

1. Jiang, W. et al. Recognition of gut microbiota by NOD2 is essential for the homeostasis of intestinal intraepithelial lymphocytes. *J. Exp. Med.* **210**, 2465-2476 (2013).
2. Rehman, A. et al. Nod2 is essential for temporal development of intestinal microbial communities. *Gut* **60**, 1354-1362 (2011).
3. Macho, Fernandez, E. et al. Anti-inflammatory capacity of selected lactobacilli in experimental colitis is driven by NOD2-mediated recognition of a specific peptidoglycan-derived muropeptide. *Gut* **60**, 1050-1059 (2011).
4. Galluzzo, S. et al. Association between NOD2/CARD15 polymorphisms and coronary artery disease: a case control study. *Human Immunol.* **72**, 636-640 (2011).
5. Vlacil, A. K. et al. Deficiency of Nucleotide-binding oligomerization domain-containing proteins (NOD) 1 and 2 reduces atherosclerosis. *Basic Res. Cardiol.* **115**, 47 (2020).
6. El Mokhtari, N. E. et al. Role of NOD2/CARD15 in coronary heart disease. *BMC genetics* **8**, 1-8 (2007).
7. Cavallari, J. F. et al. Muramyl Dipeptide-Based Postbiotics Mitigate Obesity-Induced Insulin Resistance via IRF4. *Cell Metab.* **25**, 1063-1074.e3 (2017).

Reviewer #2 (Remarks to the Author):

In this manuscript, Gao et al. examined compounds secreted by probiotic strains currently used in clinical trials. The authors identified a common secreted compound called LPH which acted as a bi-functional peptidoglycan hydrolase. In this capacity, LPH generated muramyl dipeptide which activated NOD2 and limited colitis and colitis-induced cancer. Overall, the manuscript is well written, the figures are excellent and the discussion is appropriate. A major concern is that the statistics are not properly executed. Additionally, a vehicle delivery control (pectin) should be included alone to confirm that this compound does not affect inflammation independent of LPH. Finally, some of the staining needs to be redone. However, with these changes and the other comments noted below, this work could provide the first description of the anti-colitis properties of LPH and expand our knowledge of probiotic secreted anti-inflammatory compounds.

Response to general comment: We greatly appreciate the positive remarks from the reviewer along with very helpful suggestions to improve the manuscript. We have revised the manuscript as the reviewer suggested, and please find a point-by-point response to the comments below.

Comment 1. Lactobacillus species were recently reclassified. Please update your nomenclatures as follows: Limosilactobacillus reuteri; Lacticaseibacillus casei; Lacticaseibacillus paracasei; Lacticaseibacillus rhamnosus.

Reply 1. Thanks for this suggestion, we have updated the nomenclatures accordingly.

Comment 2. Does Figure 1D depict SEM or Stdev? I would expect far more variation in the TNBS colitis in terms of weight. Additionally, the word TNBS in the figure legend of Figure 1D is cut off.

Reply 2: We used SEM in **Figure 1D** in the original article. After correcting it to Stdev, the variation in weight is bigger. Also, we checked all of the SEM or Stdev in the revised manuscript, and used Stdev uniformly in the revised manuscript. The word TNBS in the figure legend has corrected.

Comment 3. Does the pectin delivery system affect the colitis at all? The colitis models should be repeated with the pectin delivery system (without the LPH) to confirm that the delivery vehicle is not influence the inflammation. These are important controls that should be included.

Reply 3: As described in the Method section, pectin/zein beads containing BSA were prepared simultaneously with LPH and used as a control. This control was used in all of the mice gavaged with pectin/zein beads containing LPH, to avoid the impact of pectin delivery system. This protein delivery system was applied according to previous studies¹⁻².

References:

1. Yan, F. et al. Colon-specific delivery of a probiotic-derived soluble protein ameliorates intestinal inflammation in mice through an EGFR-dependent mechanism. *J. Clin. Invest.* **121**, 2242-2253 (2011).
2. Liu, L. S. et al. Pectin/zein beads for potential colon-specific drug delivery: synthesis and in vitro evaluation. *Drug Delivery* **13**, 417-423 (2006).

Comment 4. The X axis is missing from several graphs in Figure 1 and 2 (for example Figure 1E, F, H, J, L....). Please modify the graphs with the titles underneath the x-axis.

Reply 4: Thanks for the suggestions to make the figure more readable. We have added titles underneath the X-axis according to your suggestions. To make the figures concise and readable, we added group legend to the corresponding figures and annotated it in the figure legend.

Comment 5. The staining for B-defensin-2 appears to be non-specific. I recommend repeating the staining with immunostaining (fluorescence) and providing an inset with a close-up view of the tissue. Please include scale bars in the images as well.

Reply 5: We thank the reviewer for this insightful comment. Realizing that the changes of beta-defensin in our study may represent the presence or absence of functioning colonic epithelial layer, not a direct effect of LPH. We thus removed the results of defensin immunohistochemical staining. Instead, we have detected the expression of alpha-defensins (Defa-3, Defa-5, Defa-6, Defa-20) in **unaffected mice** treated with or without LPH as **reviewer 1** suggested, because alpha-defensins are the main class of paneth cells-derived antimicrobial peptides regulated by NOD2 reported

in CD patients¹⁻³. **As shown in Author Response Figure 2**, LPH has no effect on alpha-defensin expression in the **unaffected intestinal mucosa**. This result is not surprising, as there are many controversial findings about the role of the NOD2 in α -defensin expression. Simms et al. reported that decreased α -defensin expression is a consequence of inflammation per se and unrelated to the NOD2 polymorphic state, implying that altered antimicrobial function might be secondary to active CD rather than a contributor to intestinal inflammation⁴. Also, Shanahan et al. found that Nod2 does not directly regulate Paneth cell antimicrobial activity in mice⁵. The biochemical basis for this dichotomous property of Nod2 function remains unknown. Accordingly, we revised the statement about the α -defensins in the revised manuscript, as follows:

Results (Line 214-217): “secondly, LPH protects the intestinal barrier function, as shown reduced diffusion of FITC-dextran from intestinal mucosa to serum in **Fig. 5c**; thirdly, LPH could restore the dysbiotic gut microbiota induced by TNBS (**Fig. 5j-5l, Supplementary Table 2**).”

References:

1. Ayabe, T. et al. Secretion of microbicidal alpha-defensins by intestinal Paneth cells in response to bacteria. *Nat. Immunol.* **1**, 113-118 (2000).
2. Wehkamp, J. et al. NOD2 (CARD15) mutations in Crohn's disease are associated with diminished mucosal alpha-defensin expression. *Gut* **53**, 1658-64 (2004).
3. Bevins, C. L., Stange, E. F. & Wehkamp, J. Decreased Paneth cell defensin expression in ileal Crohn's disease is independent of inflammation, but linked to the NOD2 1007fs genotype. *Gut* **58**, 882-883 (2009).
4. Simms, L. A., Doecke, J. D., Walsh, M. D., Huang, N., Fowler, E. V. & Radford-Smith, G. L. Reduced alpha-defensin expression is associated with inflammation and not NOD2 mutation status in ileal Crohn's disease. *Gut* **57**, 903-910 (2008).
5. Shanahan, M. T. et al. Mouse Paneth cell antimicrobial function is independent of Nod2. *Gut* **63**, 903-910 (2014).

Comment 6. Relative optical density does not make sense for a y-axis in Figure 1J. Since the data is quantified from the images, it y-axis should say something like “relative staining intensity”.

Reply 6: We thank the reviewer for this insightful comment and agree that “relative staining intensity” is more appropriate than “relative optical density”. As mentioned

in the above (**Reply 5**), we have removed the results of defensin immunohistochemical staining, these data were not provided accordingly.

Comment 7. Based on the histology, weight and colon length, I would not say that LPH ameliorated the colitis. The histology in particular indicates that there is still colitis in LPH treated mice, just not to the same extent as the untreated mice. It would be safer to say that LPH moderately reduced the colitis.

Reply 7: Thanks for the suggestion on the choice of words to make the article more rigorous, we revised the statement as “alleviated the colitis”, which are marked with red in the revised manuscript.

Comment 8. Since only 2 doses were selected, it is inappropriate to say that there was a dose dependent response in line 103. I recommend removing this statement.

Reply 8: We have removed this statement as your suggestion.

Comment 9. The Occludin staining in Figure 5J is non-specific. I recommend doing immunostaining instead and providing an inset with a close-up view of the tissue. And rename the y-axis of the Figure 5K.

Reply 9: We thank the reviewer for this good comment. Just like the expression of defensin mentioned above, the changed expression of occludin is also a consequence of inflammation, represent the presence or absence of a functioning colonic epithelial layer, not a direct effect of LPH. **We thus removed this result from revised manuscript.** Instead, we detect the transcriptional expression of occludin in mice treated with or without LPH. As shown in **Author Response Figure 3**, LPH treatment has no effect on transcriptional expression of *Ocln* gene in **unaffected colonic mucosa**. This result indicates that the enhanced expression of occludin in LPH+TNBS mice is likely to be explained by the anti-inflammatory effect of LPH.

Author Response Figure 3. Transcriptional expression of occludin in ileal tissue of mice. C57BL/6 mice were gavaged with pectin/zein beads containing BSA or LPH (5 mg/kg body weight) for three days. Colonic tissues were harvested for qPCR analysis of *Ocln* gene. Each dot indicates an individual mouse (n = 6, female). Data are representative of 2 independent experiments and presented as mean \pm SD. ns: not significant, analyzed by Student T-test.

Comment 10. The body weight changes need to be analyzed by a repeated measures ANOVA (since they are taken over time) – not a 2-Way ANOVA. Please rerun the statistics for these types of data.

Reply 10: We thank the reviewer for this insightful comment, and we have re-performed the statistical analysis for all of these types of data. The statistical analysis of repeated measures ANOVA was added in the Methods Section of the revised manuscript. **(Line 600-604)**

Comment 11. Additionally, all the WT and NOD2KO mouse data needs to be analyzed by a 2-WAY ANOVA rather than a One Way ANOVA.

Reply 11: All the WT and *Nod2*^{-/-} mouse data were re-analyzed by a 2-WAY ANOVA accordingly. The statistical analysis of 2-WAY ANOVA on WT and NOD2KO mouse data was added in the Methods Section of the revised manuscript. **(Line 600-604)**

Reviewer #3 (Remarks to the Author):

A few queries need to be addressed:

Comment 1. How did the authors derive to the statement line no. 86 and 87". The representative sequence .. of this cluster...

Reply 1: We cluster all the 364 predicted probiotic secreted proteins at 75% identity using CD-hit (v4.8.1), and found one cluster (**Cluster 97, Supplementary Table 1**) encompassed proteins from the most of 10 clinical-established probiotics (8/10). The representative sequence, which was identified by CD-hit based on length of sequence in this cluster (the longest), was a protein from *L. casei* annotated as “hydrolase” in NCBI, without experimental validated functions.

Comment 2: How did the author obtain LPH? chromatography.. the details missing

Reply 2: We obtained LPH using recombinant expression systems, which has been described in the Methods section (**Line 351-368**), as follows:

“A schematic of the LPH structure was showed in Fig. 2a. The Sangon Biotech (Shanghai, China) was entrusted to synthesize the following gene sequences: LPH without signal sequence (51-351 aa); D316A mutation of LPH (LPH-AS1); D329A mutation of LPH (LPH-AS2); P295A, G297A and T298A mutations of LPH (LPH-AS3); LPH-3D (51-267 aa of LPH); Saga without signal sequence (51-530 aa), LRP without signal sequence (51-357 aa) and LPP without signal sequence (51-339 aa). All these sequences contained restriction enzyme sites EcoR I at the 5' end and Xho I at the 3' end. These synthetic constructs were ligated into EcoR I- and Xho I digested pET-28a respectively, and transformed into competent E. coli BL21(DE3). During inducing protein expression, kanamycin was used at 50 µg/mL, and isopropyl-β-d-thiogalactoside (IPTG) was used at 0.5 mM for LPH, LPH-AS1, LPH-AS2, LPH-AS3, LPH-3D, LRP and LPP, and at 1 mM for Saga. After induction of protein expression for 6 h, bacteria were pelleted and lysed ultrasonically. His-tagged proteins were purified using His Tagged Protein Purification Kit and desalted using phosphate buffered saline (PBS) containing 20% glycerin, followed by a clearing of potential endotoxin by passing through a Detoxi-Gel Endotoxin Removing Gel.”

Comment 3: Data of only 3D domain and peptidoglycan hydrolytic activity is missing.

Reply 3: We thank for this suggestion to help us understand the hydrolytic activity of LPH more clearly. In the original manuscript, through active site mutants and 3D-domain truncation mutation of LPH, we have convincingly demonstrated that the 3D-domain and conserved active site residues (D316, D329, G197, P295, T298) were required for LPH to generate NOD2 ligands. We agree with the reviewer that investigating the hydrolytic effects of protein with only 3D-domain will help us understand the hydrolytic activity of LPH more clearly. Thus, we recombinant the protein with only the 3D-domain (268-351, 84AA), and found that 3D domain loses the ability to generate NOD2 ligands from intact or mutanolysin-predigest peptidoglycan (**Author Response Figure 4**). This result indicates that other sequences outside the 3D-domain may help to form a complete three-dimensional protein structure.

Author Response Figure 4. 3D-domain (3D) could not generate NOD2 ligands from intact or mutanolysin-predigest peptidoglycan. NF- κ B activation in NOD2 reporter cells after incubation with BSA, LPH or 3D-domain-treated peptidoglycan. NOD2 agonist (MDP) served as the positive control. The values are ratios of cells with corresponding treatment to those with BSA treatment. Data are representative of 3 independent experiments and presented as mean \pm SD.

Comment 4: LPH shows wide substrate specificity. The products formed may differ. therefore, author needs to discuss in detail. Domain interaction study

Reply 4: We thank the reviewer for this insightful comment. Indeed, according to our study, LPH could act on peptidoglycan from *E. coli*, *S. aureus*, and *M. luteus*, while the main products are all pointed to be MDP. This could be explained by the relatively conserved structure of peptidoglycan from different bacteria, which formed from the linear chains of two alternating amino sugars, namely N-acetylglucosamine (GlcNAc) and N-acetylmuramic acid (MurNAc). Each MurNAc is attached to a short (4- to 5-residue) amino acid chain, containing L-alanine, D-glutamic acid at the first two amino acids, while the other three amino acid may vary according to different species. Since LPH catalyze the β -(1,4)-glycosidic bond at the sugars chains, and the bond between the second and third amino acid, its main product is the conserved MDP structure.

As for the “Domain interaction study” mentioned by the reviewer, only one domain was found by searching the InterPro (PFAM) and CDD databases (through NCBI). Our results indicating that some amino acids outside the 3D-domain may help to form a complete three-dimensional protein structure (**Author Response Figure 4 in Reply 3**). The crystal structure of the whole length of LPH needed to be determined in future studies to understand the hydrolytic mechanism of LPH more clearly.

Comment 5: Secondary str prediction to the level of domain is essential

Reply 5: We predict the secondary structure using web tool LambdaPP pipeline (<https://embed.predictprotein.org/o>). This result was added as Supplementary Figure 5 in the revised manuscript, see also **Author Response Figure 5** below. We found that the N-terminal (1-143) and C-terminal of this protein is composed of Helix and Sheet (231-351), while the middle of protein is mainly Helix (144-230). Two conserved aspartic acids (D316, D329) were predicted as binding sites, similar to the results of three-dimensional structure analysis (**Figure 3e, left panel**). We added this analysis to the Results section as **Supplementary Figure 5** in the revised manuscript. (**Line 148-149**); (**Line 977-982**)

Author Response Figure 5. The predicted secondary structure of LPH.

Comment 6: Individual domains of LPH cloning, and functions and their cooperative functions needs to be checked. Whether there is a inter domain interaction?

Reply 6: We thank the reviewer for this comment. We found only one domain, namely 3D-domain in LPH by searching the InterPro (PFAM) and CDD database. We have recombinant the 3D-domain, and found that 3D-domain lost the ability to generate NOD2 ligands from intact or mutanolysin-predigest peptidoglycan (**Author Response Figure 4 in Reply 3**). Our results indicating that some amino acids outside the 3D-domain may help to form a complete three-dimensional protein structure. Due to the lack of homologues on sequence out of the 3D domain, it is hard to predict the three-dimension structure of the whole length of LPH. The crystal structure of the whole length of LPH needed to be determined in future studies to understand the hydrolytic mechanism of LPH more clearly.

Comment 7: Which domain of LPH protects Mice from colitis is missing

Reply 7: Through constructing a series of active site mutants and 3D-domain truncation mutation of LPH, in combination with TNBS, DSS and OXA-induced colitis mouse models and *Nod2* knockout mice, we have convincingly demonstrated that the 3D-domain and conserved active site residues (D316, D329, G197, P295, T298) were required for LPH-mediated anti-colitis effects through generating NOD2 ligands. We guess you mean if the 3D-domain is enough to generate NOD2 ligands and protect mice from colitis. As our data shows, protein with only 3D-domain lost the ability to generate NOD2 ligands (**Author Response Figure 4 in Reply 3**). Taken that the present data have provided enough and solid evidence showing that LPH protect mice from colitis through generating NOD2 ligands, the protein with only 3D-domain is unlikely to protect mice from colitis, and we avoid repeating the animal models taken into the ethical considerations.

Comment 8: The author would have considered *Lactobacillus* as a control for LPH

Reply 8: Our recent publications, as well as quantities of other studies have shown that the LPH-encoding *Lactobacillus* could protect mice from DSS and TNBS-induced colitis, which is also our starting point¹⁻⁴. We would not consider *Lactobacillus* as a control for LPH, because the dosage of LPH (protein) and *Lactobacillus* (probiotics) are not comparable to each other.

References:

1. Hou, Q. et al. *Lactobacillus* accelerates ISCs regeneration to protect the integrity of intestinal mucosa through activation of STAT3 signaling pathway induced by LPLs secretion of IL-22. *Cell Death Differ.* **25**,1657-1670 (2018).
2. Yan, F. et al. Colon-specific delivery of a probiotic-derived soluble protein ameliorates intestinal inflammation in mice through an EGFR-dependent mechanism. *J. Clin. Invest.* **121**, 2242-53 (2011).
3. Singh, A. K., Hertzberger, R.Y. & Knaus, U. G. Hydrogen peroxide production by lactobacilli promotes epithelial restitution during colitis. *Redox Biol.* **16**, 11-20 (2018).
4. Martín, R. et al. Using murine colitis models to analyze probiotics-host interactions. *FEMS Microbiol. Rev.* **41**, S49-S70 (2017).

Comment 9: What is the amount of production of secretion of LPH by *Lactobacillus* spp how do they differ?

Reply 9: It is hard to exactly quantify the production of LPH in the supernatant of *Lactobacillus* spp using our LPH polyclonal antibody. We have attempted to semi-quantify the secretion of LPH using immunoblotting and the coomassie blue staining of total protein. The bands of coomassie blue staining were used as loading control for semi-quantitative analysis as described in our previous work and others¹⁻⁴. As shown in **Author Response Figure 6**, the amount of secreted LPH is ranked from high to low as *L. casei* > *L. paracasei* > *L. rhamnosus*. Because this result is only a semi-quantitative data, and the core of our study is focused on the LPH function, not the differences between these probiotics, we thus have not added this result in the revised manuscript.

Author Response Figure 6. Semi-quantitative analysis of LPH in the lysis and supernatant of *L. casei*, *L. paracasei* or *L. rhamnosus*. a Immunoblotting (upper panel) and corresponding coomassie blue staining of total protein (lower panel) of LPH in lysis or supernatant of probiotics. b Semi-quantitative analysis of LPH based on the internal reference of coomassie blue staining, analyzed by ImageJ. Data are representative of 3 independent experiments and presented as mean ± SD.

References:

1. Nie, X., Li, C., Hu, S., Xue, F., Kang, Y. J. & Zhang, W. An appropriate loading control for western blot analysis in animal models of myocardial ischemic infarction. *Biochem. Biophys. Res. Commun.* **12**, 108-113 (2017).
2. He, X. et al. Endogenous $\alpha 7$ nAChR Agonist SLURP1 Facilitates *Escherichia coli* K1 Crossing the Blood-Brain Barrier. *Front. Immunol.* **12**, 745854 (2021).

- Gilda, J. E. & Gomes, A. V. Stain-Free total protein staining is a superior loading control to β -actin for Western blots. *Analy. Bioche.* **440**, 186-188 (2013).
- Colella, A. D. et al. Comparison of Stain-Free gels with traditional immunoblot loading control methodology. *Anal. Biochem.* **430**, 108-110 (2012).

Comment 10: Commercial MDP should have been the best control.

Reply 10: There are many studies have shown that the commercial MDP could protect mice from DSS and TNBS-induced colitis¹⁻³. We didn't use commercial MDP as direct control for LPH because the dosage of LPH (protein) and MDP (small molecule) are not comparable to each other.

References:

- Watanabe, T. et al. Muramyl dipeptide activation of nucleotide-binding oligomerization domain 2 protects mice from experimental colitis. *J. Clin. Invest.* **118**, 545-559 (2008).
- Corridoni, D. et al. Dysregulated NOD2 predisposes SAMPI/YitFc mice to chronic intestinal inflammation. *Proc. Natl. Acad. Sci. USA* **110**, 16999-17004 (2013).
- You, Y. et al. Postbiotic muramyl dipeptide alleviates colitis via activating autophagy in intestinal epithelial cells. *Front. Pharmacol.* **13**, 1052644 (2022).

Comment 11: LPH does it maintain Eubiosis. Does it overcome the problem of Dysbiosis. Microbiota data is essential.

Reply 11: Yes, LPH restored the gut dysbiosis induced by TNBS, as shown in **Revised Manuscript Figure 5j-5l, Supplementary Table 2** (Also showed below, **Author Response Figure 7**). Furthermore, we found this effect is dependent on NOD2.

Author Response Figure 7. LPH restored the gut dysbiosis induced by TNBS. a-c Fecal 16S rRNA microbial analysis: the α -diversity reflected by Chao index (**a**), the β -diversity calculated from the operational taxonomic unit (**b**), and the bacterial composition change at the family level (**c**).

Comment 12: Lane no- 191-192. Both activities are needed for LPH protection effect. But DL-endopeptidase function is more... required why? Rationale

Reply 12: We have described the rationale already in the original manuscript (**Line 178-187**), as follows: “*To further investigate if both the DL-endopeptidase and N-Acetyl- β -D-muramidase activities were required for LPH’s colitis protective effects, we treated colitis mice with LPH, LPH-AS1/LPH-AS2, or LPH-AS3. The results showed that at a lower dose (1 mg/kg body weight), none of the LPH-AS1, LPH-AS2, or LPH-AS3 could protect mice from colitis (Supplementary Figure 6a-6e). Then we tested if these results were consistent at a higher dose of enzymes (5 mg/kg body weight). Surprisingly, both LPH-AS1 and LPH-AS2, but not LPH-AS3, exerted colitis protective effects at a higher dose (Figure 4A-4E). These results reveal that while both the DL-endopeptidase and N-Acetyl- β -D-muramidase activities are needed for LPH’s protective effect, the DL-endopeptidase function is more necessarily required.*”

Our recent public work in CHM indicated that in the complex ecology of the gut, muramidase may be a redundant enzyme in the microbiome, while the presence and abundance of DL-endopeptidase-encoding genes vary¹. Together, these data indicated that both the DL-endopeptidase and N-Acetyl- β -D-muramidase activities are needed for LPH’s protective effect, the DL-endopeptidase function is more necessarily required.

References:

1. Gao, J. et al. Gut microbial DL-endopeptidase alleviates Crohn’s disease via the NOD2 pathway. *Cell Host Microbe*, **30**, 1435-1449 (2022).

Comment 13: P40 and LPH both functions on NOD2 pathway. Are they not competitors?

Reply 13: Except for LPH, previous studies found a secreted protein P40 from several lactobacillus strains (*Lactobacillus rhamnosus* GG and *L. casei*), which could also protect mice from colitis¹. Though the protective effects from the host perspective have been well established, including activation of epidermal growth factor receptor,

regulating intestinal epithelial cell survival and growth, or promoting IgA production²⁻⁴, the direct mechanism from the protein perspective has not been established. Other studies showed that P40 is a γ -D-Glutamyl-L-Lysyl-Endopeptidase⁵⁻⁶, which is critical to generating MDP, suggesting P40 may act directly through the NOD2 pathway. If it is this way, they may act in synergy to shed NOD2 ligands to protect gut homeostasis. In addition to these important probiotic proteins, another study found that the probiotic species *L. salivarius* also protects against colitis through the NOD2 pathway⁷. Altogether, these studies suggest that NOD2 is an important colitis protective mechanism of traditional probiotics.

References:

1. Yan, F. et al. Colon-specific delivery of a probiotic-derived soluble protein ameliorates intestinal inflammation in mice through an EGFR-dependent mechanism. *J. Clin. Invest.* **121**, 2242-2253 (2011).
2. Wang, Y. et al. An LGG-derived protein promotes IgA production through upregulation of APRIL expression in intestinal epithelial cells. *Mucosal. Immunol.* **10**, 373-384 (2017).
3. Yan, F., Cao, H., Cover, T. L., Whitehead, R., Washington, M. K. & Polk, D. B. Soluble proteins produced by probiotic bacteria regulate intestinal epithelial cell survival and growth. *Gastroenterology* **132**, 562-575 (2007).
4. Regulski, K. et al. Analysis of the peptidoglycan hydrolase complement of *Lactobacillus casei* and characterization of the major γ -D-glutamyl-L-lysyl-endopeptidase. *PLoS One* **7**, e32301 (2012).
5. Bäuerl, C. et al. P40 and P75 are singular functional muramidases present in the *Lactobacillus casei/paracasei/rhamnosus* taxon. *Front. Microbiol* **10**, 1420 (2019).
6. Fernandez, E. M. et al. Anti-inflammatory capacity of selected lactobacilli in experimental colitis is driven by NOD2-mediated recognition of a specific peptidoglycan-derived muropeptide. *Gut* **60**, 1050-1059 (2011).

Comment 14: Rationale is required for Fig.O. The bacterial composition change at the family level (O).

Reply 14: Original Figure 5O (also showed as **Author Response Figure 8 below**) shows the bacteria (at family level) that are significantly changed by TNBS, and could be restored by LPH dependent on NOD2. The bubble size of WT-Con was set at 1, and the size indicates the mean relative change to WT-Con.

Author Response Figure 8. Change of bacterial composition at the family level.

Comment 15: Why is the Bifido content gets reduced after treatment with LPH and maintaining enterobacteriace constant.

Reply 15: As shown in the **Original Figure 50** (also showed as **Author Response Figure 8 above**), in WT mice, treatment with TNBS reduced Bifidobacteriaceae abundance compared with control, while added LPH increased Bifidobacteriaceae content (**red** solid bubble) and reduced Enterobacteriaceae (**blue** solid bubble). However, in *Nod2*^{-/-} mice, the Bifidobacteriaceae or Enterobacteriaceae content is not different between TNBS and LPH+TNBS groups. These results indicate that LPH regulate gut microbiota dysbiosis through NOD2 pathway.

The above reasons may make the MS effective therefore, they may be considered to look in very seriously.

Reply 16: We greatly appreciate these comments from the reviewer along with very helpful suggestions to improve the manuscript. We have discussed all the comments above.

REVIEWER COMMENTS

Reviewer #1 (Remarks to the Author):

The authors have satisfactorily responded to most of my comments. I am still puzzled by the beta diversity plot, which now shows that in WT animals TNBS has virtually no effect (green) while co-treatment of WT with DSS and LPH makes the animals more similar to the NOD2/TNBS treated group. The bubble plot cannot be understood from figure/legend, as the size of the bubbles is just given as 1,2, 5, 10 and the legend states "bacterial composition change at the family level" , there is no explanation on statistics or unit of change. This must be corrected and explained, so that the reader can judge whether the indicated taxa represent a statistically significant effect.

Reviewer #2 (Remarks to the Author):

The authors have done an excellent job addressing any initial concerns.

Reviewer #3 (Remarks to the Author):

1. Both activities are needed for LPH protection effect.

But DL-endopetidase function is more... required why? Rationale? The authors answer for the above query is not convincing.

2. LPH shows wide substrate specificity. The products formed may differ.therefore, author needs to discuss in detail. Domain interaction study

Ans: author response is not convincing

3.LPH knockout will further enhance the authors claims.

4. LPH (protein) and MDP (small molecule) are not comparable to each other. if the function of the molecules are identical there is nothing wrong in considering MDP as a control.

5. Individual domains of LPH cloning, and functions and their cooperative functions needs to be checked. Whether there is a inter domain interaction?

answer: Author response is not convincing

REVIEWER COMMENTS

Reviewer #1 (Remarks to the Author):

The authors have satisfactorily responded to most of my comments. I am still puzzled by the beta diversity plot, which now shows that in WT animals TNBS has virtually no effect (green) while co-treatment of WT with DSS and LPH makes the animals more similar to the NOD2/TNBS treated group.

The bubble plot cannot be understood from figure/legend, as the size of the bubbles is just given as 1,2, 5, 10 and the legend states "bacterial composition change at the family level", there is no explanation on statistics or unit of change. This must be corrected and explained, so that the reader can judge whether the indicated taxa represent a statistically significant effect.

Response: We sincerely appreciate your careful review of our manuscript, and are pleased to hear that you found our responses satisfactory for most of your comments. Regarding the beta diversity plot, we apologize for the mistake in the figure legend, which has now been corrected as follows (**Author Response Figure 1 c, Revised Manuscript Figure 5 k**):

Author Response Figure 1: Microbiota 16S rRNA analysis of WT/*Nod2*^{-/-} mice. (a) Figure 5K in the original manuscript. (b) the bar plot of values on axis one of the beta-diversity PCoA plot. (c) Replotted PCoA in the revised manuscript.

To improve the clarity of the bubble plot, we have changed it to a heatmap format (**Author Response Figure 2, Revised Manuscript Figure 5 I**). The rows indicate each treatment group, while the columns indicate the mean relative abundance of bacterial taxa at the family level. The data have been normalized according to column (L1 normalization). Only bacteria that were significantly altered by TNBS in WT mice and could be restored by LPH depending on NOD2 are displayed. We have also included a

more detailed explanation of the statistical analysis in the Figure legend (Line 899-904).

Author Response Figure 2: The mean abundance of bacterial composition at the family level according to different groups, all the data are normalized according to column (L1 normalization). Only bacteria that are significantly changed by TNBS in WT mice, and could be restored by LPH dependent on *Nod2* are displayed: * $p < 0.05$ compared with WT-Con, + $p < 0.05$ compared to WT-TNBS, # $p < 0.05$ compared to *Nod2*^{-/-}-Con.

Reviewer #2 (Remarks to the Author):

The authors have done an excellent job addressing any initial concerns.

Response: We are pleased to hear that our revisions have addressed any initial concerns of the reviewer. We have worked diligently to ensure that our manuscript meets the high standards of *Nature Communications* and provides valuable insights into the role of probiotics in regulating gut homeostasis.

Reviewer #3 (Remarks to the Author):

1. Both activities are needed for LPH protection effect. But DL-endopeptidase function is more... required why? Rationale? The authors answer for the above query is not convincing.

Response 1: We would like to thank the reviewer for taking the time to review our manuscript, and for continued engagement with our work. The statement “*While both the DL-endopeptidase and N-Acetyl-β-D-muramidase activities are needed for LPH’s protective effect, the DL-endopeptidase function is more necessarily required*”

in the original article is conclude from the experiments that the mutants with only DL-endopeptidase activity (LPH-AS1/LPH-AS2) and the mutant with only the N-Acetyl- β -D-muramidase activity (LPH-AS3) loss the colitis protective effects at low dose, while at higher dose, mutants with only DL-endopeptidase activity, rather than those with N-Acetyl- β -D-muramidase activity restore the colitis protective effects.

After carefully considered feedback from the reviewer, we think the above conclusion is not vigor enough with regard to the current experiments, and we have adapt the statements from: “*These results reveal that while both the DL-endopeptidase and N-Acetyl- β -D-muramidase activities are needed for LPH’s protective effect, the DL-endopeptidase function is more necessarily required*” to “*These results reveal that the DL-endopeptidase activities of LPH is critical to alleviate colitis in mice*” **Results section (Line 171-180)**. We illustrated the rational of the changed statements as follows:

We have construct LPH mutant that maintains only the DL-endopeptidase activity (LPH-AS1/LPH-AS2) or the N-Acetyl- β -D-muramidase activity (LPH-AS3). Then we treated colitis mice with LPH-AS1/LPH-AS2, or LPH-AS3. The results showed that at a lower dose (1 mg/kg body weight), neither of the LPH-AS1, LPH-AS2, or LPH-AS3 could protect mice from colitis (**Supplementary Figure 6a-6e**). However, at a higher dose of enzymes (5 mg/kg body weight), both LPH-AS1 and LPH-AS2 (with DL-endopeptidase activity), but not LPH-AS3 (N-Acetyl- β -D-muramidase activity), exerted colitis protective effects.

In the later experiments, we proved that the colitis protective effects of these mutants correlate with their ability to generate NOD2 ligands. In detail, none of these mutants could elevate NOD2 ligands at a lower dose, while both LPH-AS1 and LPH-AS2 (with DL-endopeptidase activity), but not LPH-AS3 (with N-Acetyl- β -D-muramidase activity) could elevate NOD2 ligands at a higher dose.

These findings are consistent with our recent publication in *Cell Host & Microbe*¹, which demonstrated that N-Acetyl- β -D-muramidase is a redundant enzyme in the gut microbiome, and that supplementation with N-Acetyl- β -D-muramidase (LPH-AS3) has negligible effects in shedding NOD2 ligands, which is critical for the colitis protective role of LPH. Furthermore, we found that the abundance of DL-endopeptidase-encoding genes is relatively lower than N-Acetyl- β -D-muramidase in the gut microbiome. Thus, supplementation of DL-endopeptidase (LPH-AS1/LPH-AS2) could elevate the NOD2 ligands in the gut tissue and feces, leading to colitis protective effects.

Taken together, we conclude that the DL-endoropeptidase activities of LPH is critical to alleviate colitis in mice. We have added the discussion about rational in the **Results section (Line 195-201)**, and hope that the revisions and additional explanations address the concerns of the reviewers and makes the manuscript more rigor.

References:

1. Gao, J. et al. Gut microbial DL-endoropeptidase alleviates Crohn's disease via the NOD2 pathway. *Cell Host Microbe*, **30**, 1435-1449 (2022).

Results (Line 171-180): *To further investigate the role of DL-endoropeptidase and N-Acetyl- β -D-muramidase activities in LPH's colitis protective effects, we treated colitis mice with LPH, LPH-AS1/LPH-AS2 (retain DL-endoropeptidase activity), or LPH-AS3 (retain N-Acetyl- β -D-muramidase activity). The results showed that at a lower dose (1 mg/kg body weight), none of the LPH-AS1, LPH-AS2, or LPH-AS3 could protect mice from colitis (Supplementary Fig. 6a-6e). Then we tested if these results were consistent at a higher dose of enzymes (5 mg/kg body weight). Surprisingly, both LPH-AS1 and LPH-AS2, but not LPH-AS3, exerted colitis protective effects at a higher dose (Fig. 4a-4e). These results reveal that the DL-endoropeptidase activities of LPH is critical to alleviate colitis in mice.*

Results (Line 195-201): *These findings are consistent with our recent study³⁷, which demonstrated that N-Acetyl- β -D-muramidase is a redundant enzyme in the gut microbiome, while the abundance of DL-endoropeptidase is relatively lower than N-Acetyl- β -D-muramidase. Thus, supplementation of DL-endoropeptidase (LPH-AS1/LPH-AS2), instead of N-Acetyl- β -D-muramidase (LPH-AS3) could elevate the NOD2 ligands in the gut. All these results indicate that DL-endoropeptidase activities of LPH and the generated NOD2 ligands are sufficient for LPH's colitis-protective effects.*

2. LPH shows wide substrate specificity. The products formed may differ. therefore, author needs to discuss in detail. Domain interaction study. Author response is not convincing.

Response 2: We acknowledge and appreciate the valuable feedback provided by the Reviewer concerning the broad substrate specificity of LPH and its potential to generate diverse peptidoglycan products. As reported in our study, we have observed that LPH can generate NOD2 ligands from both intact and mutanolysin-predigested peptidoglycan derived from *E. coli*, *S. aureus*, and *M. luteus*, and have employed NOD2-reporter cells, ANTS labeling, and HPLC-MS/MS to characterize these products. However, we recognize that our analysis may have overlooked other products aside from NOD2 ligands. The distinct peptidoglycan structures across Gram-positive and Gram-negative bacterial species, including differences in amino acid chains and cross-linked bridges, can potentially lead to the generation of additional products that are dependent on the peptidoglycan architecture of the specific bacteria. While our study did not fully characterize the diverse peptidoglycan products produced by LPH from different bacterial species, we have acknowledged this limitation in the **Discussion section** of the revised manuscript (**Lines 294-301**).

Regarding the "Domain interaction study" raised by the reviewer, we have provided a careful response to this inquiry in **Response 5**.

Discussion (Lines 294-301):

The present study has several limitations that should be noted. Firstly, the wide substrate specificity of LPH enables it to generate NOD2 ligands from diverse bacterial sources. However, the amino acid composition of the oligopeptide chain and the cross-linked bridges within the peptidoglycan can vary significantly among different bacteria, which may lead to the production of alternative products that differ based on the peptidoglycan structure. The precise characterization of these diverse peptidoglycan products from various bacterial species was not accomplished in our study, and as such, further investigations are required to elucidate the potential of products beyond MDP.

3.LPH knockout will further enhance the authors claims.

Response 3: Thank the reviewer for this insightful comment. We acknowledge that an LPH knockout strain could potentially serve as additional evidence to confirm the role of LPH as the primary effector of the observed effects in these *Lactobacillus* strains. However, modifying the genome of *Lactobacillus*, which is a Gram-positive bacterium, poses significant technical challenges that are beyond the scope of our current study. Our research, nevertheless, presents robust evidence for the involvement of LPH in gut homeostasis regulation, as exemplified by the findings of multiple experiments and analyses. Nonetheless, it is plausible that other effectors from *Lactobacillus* could contribute to the observed protective effects against colitis, as previously suggested by the discovery of P40 and P75¹. We appreciate your valuable suggestion and will consider the feasibility of generating an LPH knockout strain in future studies. We have also included a discussion of this possibility in our manuscript's revised version (Lines 279-281).

References:

1. Yan, F. et al. Colon-specific delivery of a probiotic-derived soluble protein ameliorates intestinal inflammation in mice through an EGFR-dependent mechanism. *J. Clin. Invest.* **121**, 2242-2253 (2011).

Discussion (Lines 279-281):

Additionally, the need for specific gene knock-out lactobacillus strains is imperative to elucidate the primary effector(s) responsible for the observed colitis-protective effects of these probiotic strains.

4. LPH (protein) and MDP (small molecule) are not comparable to each other. if the function of the molecules are identical there is nothing wrong in considering MDP as a control.

Response 4: We express our gratitude for this insightful comment, and we have incorporated additional experiments in response, as follows: We designed an experiment to evaluate and compare the protective effects of LPH and MDP on TNBS-induced colitis. Mice were gavaged with a various dose of LPH (1-5 mg/kg) or MDP (2-10 mg/kg) for three days. Our results, presented in the **Author Response Figure 3 (Revised manuscript supplementary Figure 6f-6l)**, indicate that LPH (2.5 mg/kg)

and MDP (5 mg/kg) yielded comparable increases in NOD2 ligands in both colon and fecal samples. Moreover, we found that both LPH and MDP exhibited comparable protective roles against TNBS-induced colitis at these doses, as demonstrated by the data presented in the **Author Response Figure 3 c-g**. We have incorporated these findings into the results section (Lines 191-195) and **Supplementary Figure 6** of revised manuscript.

Author response Figure 3: Protective Effects of LPH and MDP on TNBS-Induced Colitis. Mice were treated with pectin/zein beads containing LPH (1-5 mg/kg) or MDP (2-10 mg/kg) before being challenged with TNBS for 4 days. Pectin/zein beads containing BSA were administered into Con and MDP mice to avoid the effect of Pectin/zein. The ability of colonic homogenate (**a**) or fecal extract (**b**) to activate NOD2-expressing NF- κ B reporter HEK293 cells were detected. Data are presented as fold change relative to BSA treated group (Con). **c-g** The protective effects of LPH and MDP on colitis were evaluated by measuring body weight loss (**c**), colon length (**d**), serum FITC-dextran level (**e**), and semiquantitative scoring of inflammation (**g**). Representative H&E staining of colonic tissue was also performed, and the scale bar is 200 μm (**f**). Each dot indicates an individual mouse (n = 6, female). Data are representative of 3 independent experiments and presented as mean \pm SD. * $p < 0.05$, ** $p < 0.01$, *** $p < 0.001$. ns: not significant.

Results (Lines 191-195):

Moreover, we validated that MDP, a well-established NOD2 ligand, displayed equivalent protective effects against TNBS-induced colitis at a dosage of 5 mg/kg in comparison to LPH at 2.5 mg/kg, which correspondingly resulted in analogous elevation of NOD2 ligands in both fecal and colonic samples (**Supplementary Figure 6f-6l**).

5. Individual domains of LPH cloning, and functions and their cooperative functions needs to be checked. Whether there is a inter domain interaction? Author response is not convincing.

Response 5: We appreciate the thoughtful comment from the reviewer regarding the significance of understanding the interactions between the two domains of LPH. As a response to this comment, we utilized AlphaFold to predict the 3-dimensional structure of the full-length protein, which revealed two domains at the N-terminal (44D-116P) and C-terminal (268F-350I) regions with relatively high confidence. However, the sequence between these two domains (117-262) was predicted with low confidence (**Author response Figure 4 a-b**). To investigate the role of the individual domains, we conducted *in-vitro* experiments using recombinant proteins and found that neither domain was capable of generating NOD2 ligands from intact or mutanolysin-predigested peptidoglycan (**Author response Figure 4 c**). Additionally, we conducted docking studies of each domain with MurNAc-pentapeptide using Autodock4, which did not indicate stable ligand-bound poses in either domain, with the best docked energies ranging from 3 to 6 (kcal/mol) (**Author response Table 1**). These findings are consistent with the experimental results (**Author response Figure 4 c**) and suggest that the interaction and cooperation between the two domains are necessary for the hydrolytic activity of LPH. Consequently, we conducted docking studies of the region between the two domains with MurNAc-pentapeptide and found a ligand-bound poses in this area, with best docked energies ranging from -0.17-2.33 (**Author response Table 1, Author response Figure 4 d**). We acknowledge that the results may not be reliable due to the low confidence in the predicted 3-dimensional structure of the region between the two domains, which could significantly impact their interactions.

Author response Figure 4. 3D-domain (3D) and 44D-116P domain could not generate NOD2 ligands from intact or mutanolysin-predigest peptidoglycan. **a** The predicted 3-dimensional structure of LPH by AlphaFold. Some regions below 50 pLDDT may be unstructured in isolation. **b** Per-residue confidence score (pLDDT) provided by AlphaFold. **c** NF- κ B activation in NOD2 reporter cells after incubation with BSA, LPH, 3D-domain or 44D-116P domain -treated peptidoglycan. NOD2 agonist (MDP) served as the positive control. The values are ratios of cells with corresponding treatment to those with BSA treatment. Data are representative of 3 independent experiments and presented as mean \pm SD. **d** The docking studies of the region between the two domains with MurNAc-pentapeptide.

Author response Table1: The docked energy between the N-terminal domain (44D-116P, LPH-N), the C-terminal domain (268F-350I, LPH-C), and the region between the two domains (LPH-Gap) of LPH with M-penta.

LPH-N (kcal/mol)	LPH-C (kcal/mol)	LPH-Gap (kcal/mol)
3.9	3.63	-0.17
4.68	4.08	0.77
4.84	4.28	0.78
5.01	4.44	0.95
5.04	4.49	1.59
5.54	4.66	1.7
5.7	5.05	2.04
5.79	5.37	2.05
5.96	5.38	2.15
6	5.5	2.33

To further provide the structural evidence of LPH's hydrolytic function, we use SWISS-MODEL to search the 3-dimensional templates of LPH, and found some protein match to the N-terminal or C-terminal regions separately, some of which are annotated as Bifunctional autolysin or membrane-bound lytic murein transglycosylase A (<https://swissmodel.expasy.org/interactive/ZgmHzr/templates/>) (**Author response Figure 5**). Though these results provide additional information about the rational of LPH's hydrolytic function, due to lack of homologues with good solved 3-dementional structure, the domain interaction is unavailable at this time. To overcome this limitation, future studies will aim to determine the crystal structure of the whole length of LPH. We have included this limitation in the Discussion section of the revised manuscript (**Lines 302-309**). We trust that this additional information addresses the reviewer's concern, and we express our appreciation for their feedback.

4wvk.1.A YuiC Stationary Phase Survival Protein YuiC from B.subtilis complexed with reaction product		0.10	0.03	32.93	X-ray, 2.0Å	homo-dimer Δ
4wjt.1.A Uncharacterized protein YuiC Stationary Phase Survival Protein YuiC from B.subtilis complexed with NAG		0.09	0.08	33.33	X-ray, 1.2Å	homo-dimer Δ
4wli.1.A YuiC Stationary Phase Survival Protein YuiC from B.subtilis		0.09	0.06	32.10	X-ray, 1.8Å	homo-dimer Δ
7kwi.1.A Bifunctional autolysin Solution Structure of the R2ab Repeat Domain from Staph. epidermidis Autolysin (AtIE)		0.06	-	15.00	NMR	monomer \checkmark
7esj.1.A membrane-bound lytic murein transglycosylase A Acinetobacter baumannii membrane-bound lytic murein transglycosylase A		0.05	-	28.00	X-ray, 2.1Å	homo-dimer Δ
4epc.1.A N-acetylmuramoyl-L-alanine amidase Crystal structure of Autolysin repeat domains from Staphylococcus epidermidis		0.05	-	15.52	X-ray, 2.9Å	monomer \checkmark

Author response Figure 5. The 3-dimensional structure templates of LPH generated by SWISS-MODEL.

Discussion (Lines 302-309)

Another limitation of the study pertains to the lack of a crystal structure for the full-length LPH protein, which could provide additional insights into the nature of inter-domain interactions and their respective contributions to LPH's biological functions. However, the task of determining the crystal structure of a protein is a challenging and resource-intensive process that may require a considerable amount of time and effort. Despite this limitation, our study has provided significant new knowledge about the functional roles of LPH, underscoring its potential as a therapeutic agent for regulating gut homeostasis.

REVIEWERS' COMMENTS

Reviewer #1 (Remarks to the Author):

Last points were addressed appropriately.

Reviewer #3 (Remarks to the Author):

As per my understanding there are no gaps existing in the MS. All the queries raised has been successfully answered and delivered in the manuscript. Therefore, the MS has addressed all the queries and the story is complete without gaps. Nice to see coming up in NC.

REVIEWER COMMENTS

Reviewer #1 (Remarks to the Author):

Last points were addressed appropriately.

Response: Thank you for your valuable comments. We are delighted to learn that the previous concerns have been adequately addressed. Your feedback is greatly appreciated, and we thank you for taking the time to review our work.

Reviewer #3 (Remarks to the Author):

As per my understanding there are no gaps existing in the MS. All the queries raised has been successfully answered and delivered in the manuscript. Therefore, the MS has addressed all the queries and the story is complete without gaps. Nice to see coming up in NC.

Response: We are pleased to receive your feedback that the manuscript has successfully addressed all the queries and there are no gaps in the story. Your thoughtful evaluation and insightful comments are highly valued. Thank you for dedicating your time to reviewing our work.